# A new dataset of river flood hazard maps for Europe and the Mediterranean Basin

**Francesco Dottori[1], Lorenzo Alfieri[2], Alessandra Bianchi[3], Jon Skoien[1], Peter Salamon[1]**

1: European Commission, Joint Research Centre, Via E. Fermi 2749, 21027 Ispra, Italy.

2: CIMA Research Foundation, Savona, Italy

3: FINCONS SPA, Italy

Correspondence to: francesco.dottori@ec.europa.eu

**Keywords:** river flooding, flood hazard mapping, Europe, EFAS, Mediterranean Basin.

## Abstract

In recent years, the importance of continental-scale hazard maps for riverine floods has grown. Nowadays, such maps are used for a variety of research and commercial activities, such as evaluating present and future risk scenarios and adaptation strategies, as well as supporting national and local flood risk management plans. In this paper we present a new set of high-resolution (100 metres) hazard maps for river flooding that covers most European countries, as well as all of the river basins entering the Mediterranean and Black Seas in the Caucasus, Middle East and Northern Africa countries. The new river flood hazard maps represent inundation along 329,000 km of the river network, for six different flood return periods, expanding on the datasets previously available for the region. The input river flow data for the new maps are produced by means of the hydrological model LISFLOOD using new calibration and meteorological data, while inundation simulations are performed with the hydrodynamic model LISFLOOD-FP. In addition, we present here a detailed validation exercise using official hazard maps for Hungary, Italy, Norway, Spain and the UK, which provides a more detailed evaluation of the new dataset compared with previous works in the region. We find that the modelled maps can identify on average two-thirds of reference flood extent, but they also overestimate flood-prone areas for flood probabilities below 1-in-100 years, while for return periods equal to or above 500 years the

maps can correctly identify more than half of flooded areas. Further verification is required in North African and Eastern Mediterranean regions, in order to understand better the performance of the flood maps in arid areas outside Europe. We attribute the observed skill to a number of shortcomings of the modelling framework, such as the absence of flood protections and rivers with upstream area below 500 $km^2$, and the limitations in representing river channels and topography of lowland areas. In addition, the different designs of reference maps (e.g. extent of areas included) affect the correct identification of the areas for the validation, thus penalizing the scores. However, modelled maps achieve comparable results to existing large-scale flood models when using similar parameters for the validation. We conclude that recently released high-resolution elevation datasets, combined with reliable data of river channel geometry, may greatly contribute to improving future versions of continental-scale river flood hazard maps. The new high-resolution database of river flood hazard maps is available for download at http://data.europa.eu/89h/1d128b6c-a4ee-4858-9e34-6210707f3c81 (Dottori et al., 2020a).

## 1) Introduction

Nowadays, flood hazard maps are a basic requirement of any flood risk management strategy (EC 2007). Such maps provide spatial information about a number of variables (e.g. flood extent, water depth, flow velocity) that are crucial to quantify flood impacts and therefore to evaluate flood risk. Moreover, they can be used as a powerful communication tool, enabling the quick visualization of the potential spatial impact of a river flood over an area.

In recent years, continental- and global-scale flood maps have grown in importance, and these maps are now used for a variety of research, humanitarian and commercial activities, and as a support of national and local flood management (Ward et al., 2015; Trigg et al., 2016). Global flood maps are used to provide flood risk information and to support decision-making in spatial and infrastructure planning, in countries where national level assessments are not available (Ward et al., 2015). Moreover, continental and global hazard maps are vital for consistent quantification of flood risk and for projecting the impacts of climate change (Alfieri et al., 2015; Trigg et al., 2016; Dottori et al., 2018), thereby allowing for comparisons between different regions, countries and river basins (Alfieri et al., 2016). Quantitative and comparable flood risk assessments are also necessary to derive measurable indicators of the targets set by international agreements such as the Sendai Framework for Disaster Risk Reduction (UNISDR, 2015).

In Europe, continental-scale flood hazard maps have been produced by Barredo et al. (2007), Feyen et al. (2012), Alfieri et al. (2014), Dottori et al. (2016a) and Paprotny et al. (2017). These maps have been used for a variety of studies, such as the evaluation of river flood risk under future socio-economic and climate scenarios (Barredo et al.,2007; Feyen et al., 2012; Alfieri et al., 2015), the evaluation of flood adaptation measures (Alfieri et al., 2016) and near real-time rapid risk assessment (Dottori et al., 2017).

The quality of continental-scale flood maps is constantly improving, thanks to the increasing accuracy of datasets and modelling tools. Wing et al., (2017) developed a dataset of flood hazard maps for the conterminous United States using detailed national datasets and high-resolution hydrodynamic modelling, and demonstrated that continental-scale maps can achieve an accuracy similar to official national hazard maps, including maps based on accurate local-scale studies. Moreover, Wing et al. used the same official hazard maps to evaluate the performance of the global flood hazard model developed by Sampson et al. (2015). While the global model was less accurate than the continental version, it was able to identify correctly over two-thirds of flood

extent. Conversely, European-scale maps have undergone limited testing against the official
hazard maps, due to limitations in accessing official data (Alfieri et al., 2014).
Here, we present a new set of flood hazard maps at 100 metres resolution (Dottori et al., 2020a),
developed as a component of the Copernicus European Flood Awareness System (EFAS,
www.efas.eu). The new dataset builds upon the map catalogue developed by Dottori et al (2016a),
and features several improvements. The geographical extent of the new maps has been expanded
to include all geographical Europe (with the exclusion of the Volga river basin), the rivers entering
the Mediterranean Sea and the Black Sea (with the partial inclusion of the Nile river basin), plus
Turkey, Syria and the Caucasus region. To the best of our knowledge, these are the first flood
hazard maps available at 100 metres resolution for the whole region of the Mediterranean Basin.
The hydrological input data are calculated using the LISFLOOD hydrological model (van der
Knijff et al., 2010; Burek et al, 2013; https://ec-jrc.github.io/lisflood/), based on updated routines
and input data in respect to the previous dataset by Dottori et al. (2016a). Flood simulations are
performed with the hydrodynamic model LISFLOOD-FP (Bates et al., 2010; Shaw et al., 2021),
following the approach developed by Alfieri et al., (2014; 2015).
To provide a comprehensive overview of the skill of the new hazard maps, we perform a
validation exercise using official hazard maps for a number of countries, regions and large river
basins in Europe. The number and extent of the validation sites allows for a more detailed
evaluation with respect to previous efforts by Alfieri et al. (2014) and Paprotny et al. (2017), even
though none of the validation sites is located outside Europe (due the unavailability of national
flood maps). Finally, we discuss the results of the validation in light of previous literature studies,
we compare the performance of the present and previous versions of the flood hazard map dataset,
and we discuss a number of tests with alternative datasets and methods.

## 97  *2) Data and methods*

In this Section we describe the procedure adopted to produce and validate the flood hazard maps.
The hydrological input data consist of daily river flow for the years 1990-2016, produced with
the hydrological model LISFLOOD (see Section 2.1), based on interpolated daily meteorological
observations. River flow data are analysed to derive frequency distributions, peak discharges and
flood hydrographs, as described in Section 2.2. Flood hydrographs are then used to simulate
flooding processes at local scale with the LISFLOOD-FP hydrodynamic model (Section 2.3).
Finally, Section 2.4 describes the validation exercise and the comparison of different approaches
and input datasets.

## 2.1 The LISFLOOD model

LISFLOOD (Burek et al, 2013; van der Knijff et al., 2010) is a distributed, physically-based
rainfall-runoff model combined with a routing module for river channels. For this work we used
an updated version of LISFLOOD, released as open-source software and available at https://ec-
jrc.github.io/lisflood/. The new version features an improved routine to calculate water
infiltration, the possibility of simulating open water evaporation and minor adjustments that
correct previous code inconsistencies (Arnal et al., 2019). The model is applied to run a long-term
hydrological simulation for the period 1990-2016 at 5 km grid spacing and at daily resolution,
which provides the hydrological input data for the flood simulations. Note that the same
simulation also provides initial conditions for daily flood forecast issued by EFAS.
The long-term run of LISFLOOD is driven by gridded meteorological maps, derived by
interpolating meteorological observations from stations and precipitation datasets (see Appendix
A for details). The meteorological dataset has been updated with respect to the dataset used by
Dottori et al. (2016a), to include new stations and gridded datasets across the new EFAS domain
(Arnal et al. 2019). In addition, LISFLOOD simulations require a number of static input maps
such as land cover, digital elevation model (DEM), drainage network, soil parameters and
parameterization of reservoirs. All the static maps have been updated to cover the whole EFAS
domain depicted in Figure 1. Further details on the static maps are provided by Arnal et al. (2019).
The current LISFLOOD version also benefits from an updated calibration at European scale,
based on the Evolutionary Algorithm approach (Hirpa et al., 2018) with the modified Kling-Gupta
efficiency criteria (KGE; Gupta et al., 2009) as objective function, and streamflow data for 1990-
2016 from more than 700 gauge stations. The same stations have been used to validate model
results, considering different periods of the time-series. The calibration and validation procedure
and the resulting hydrological skill are described by Arnal et al (2019), and summarized in
Appendix B. While we did not carry out a formal comparison with the previous LISFLOOD
calibration, which used a different algorithm and performance indicators (Zajac et al., 2013), the
larger dataset of streamflow observations and the improvement of the calibration routines should
provide a better performance.
The geographical extent used in the present study to produce the flood maps follows the recent
enlargement of EFAS (Arnal et al., 2019), and is shown in Figure 1. The new domain is
approximately 8,930,000 km$^2$ wide (an increase of 76% compared with the previous extent). The
new extent covers the entire area of geographical Europe (with the exclusion of the Volga river
basin and a number of river basins of the Arctic Sea in Russia), all the rivers entering the
Mediterranean and Black Seas (with a partial inclusion of the Nile river basin), plus the entire
territories of Armenia, Georgia, Turkey, and most of Syria and Azerbaijan.

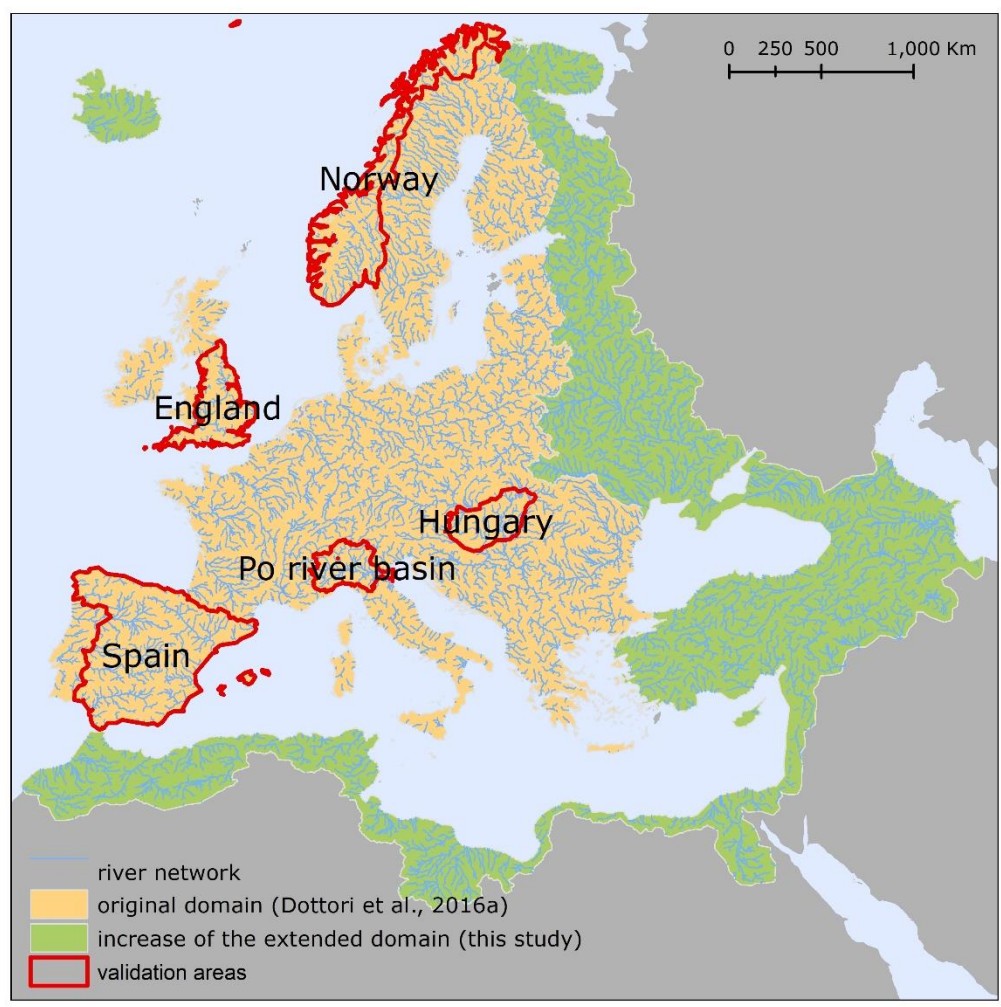


*Figure 1. Geographical extent of the EFAS extended domain covered by the present dataset of*
*flood hazard maps. The extent of the map dataset produced by Dottori et al. (2016a) is depicted*
*in beige, while the regions added with the extended domain are in green. The Figure also displays*
*the river network considered by the flood maps and the areas used for the validation exercise (see*
*Sections 2.3 and 3).*
The river network included in the new flood hazard maps has a total length of 329,000 km, with
an 80% increase compared with the previous flood maps (Alfieri et al., 2015; Dottori et al.,
2016a).

## 2.2 Hydrological input of flood simulations

The hydrological input data required for the flood simulations are provided using synthetic flood
hydrographs, following the approach proposed by Alfieri et al. (2014).
We use the streamflow dataset derived from the long-term run of LISFLOOD described in Section
2.1, considering the rivers with upstream drainage areas larger than 500 km$^2$. This threshold was
selected because the meteorological input data cannot accurately capture the short and intense
rainfall storms that induce extreme floods in small river basins, and therefore the streamflow
dataset does not represent accurately the flood statistics of smaller catchments (Alfieri et al.,

158     2014).

For each pixel of the river network we selected annual maxima over the period 1990-2016 and
we used the L-moments approach to fit a Gumbel distribution and calculate peak flow values for
reference return periods of 10, 20, 50, 100, 200 and 500 years. We also calculated the 30- and
1,000-year return periods in limited parts of the model domain to allow validation against official
hazard maps, see Section 2.3. The resulting goodness-of-fit is presented and discussed in
Appendix B. We used the Gumbel distribution to keep a parsimonious parameterization (two
parameters instead of three for the generalized extreme value (GEV), log-normal and other
distributions), thus avoiding over-parameterization when extracting high return period maps from
a relatively short time-series. The same distribution was also adopted for the extreme value
analysis in previous studies regarding flood frequency and hazard (Alfieri et al., 2014, 2015;
Dottori et al., 2016).
Subsequently, we calculate a Flow Duration Curve (FDC) from the streamflow dataset. The FDC
is obtained by sorting in decreasing order all the daily discharges, thus providing annual
maximum values $Q_D$ for any duration i between 1 and 365 days. Annual maximum values are
then averaged over the entire period of data, and used to calculate the ratios $\varepsilon_i$ between each
average maximum discharge for i-th duration $Q_{D(i)}$ and the average annual peak flow (i.e. $Q_D = 1$
day). Such a procedure was carried out for all the pixels of the river network.
The synthetic flood hydrographs are derived using daily time-steps, following the procedure
proposed by Maione et al. (2003). The peak value of the hydrograph is given by the peak discharge
for the selected T-year return period $Q_T$, while the other values for $Q_i$ are derived by multiplying
$Q_T$ by the ratio $\varepsilon_i$. The hydrograph peak $Q_T$ is placed in the centre of the hydrograph, while the
other values for $Q_i$ are sorted alternatively as shown in Figure 2. The resulting hydrograph shape
is therefore fully consistent with the empirical values of the flow duration curve. The total
duration of the synthetic hydrograph is given by the local value of the time of concentration $T_c$,
such that all of the durations $> T_c$ are discarded from the final hydrograph (Figure 2).

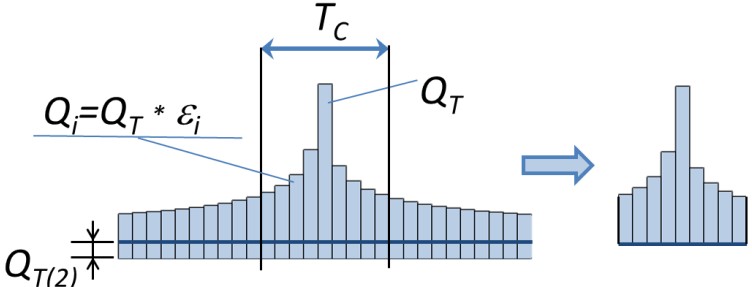


*Figure 2. General scheme of flood hydrographs (adapted from Alfieri et al., 2014).*

Because river channels are usually not represented in continental-scale topography, flood
hydrograph values are reduced by subtracting the 2-year discharge peak $Q_{T(2)}$, which is commonly
considered representative of river bank-full conditions. (Note that the original DEM is not
modified with this procedure). Hence, the overall volume of the flood hydrograph is given by the
sum of all daily flow values with duration $< T_c$.

## 2.3 Flood hazard mapping
The continental-scale flood hazard maps are derived from local flood simulations run along the
entire river network, as in Alfieri et al. (2014). We use the DEM at 100 metres resolution
developed for the Catchment Characterization and Modelling Database (CCM; Vogt et al., 2007)
to derive a high-resolution river network at the same resolution. Along this river network we
identify reference sections every 5 km along the stream-wise direction, and we link each section
to the closest upstream section (pixel) of the EFAS 5 km river network, using a partially
automated procedure to ensure a correct linkage near confluences. In this way, the hydrological
variables necessary to build the flood hydrographs can be transferred from the 5 km to the 100
metres river network. Figure 3 describes how the 5 km and 100 metres river sections are linked
using a conceptual scheme.
Then, for every 100 metres river section we run flood simulations using the two-dimensional
hydrodynamic model LISFLOOD-FP (Shaw et al., 2021), to produce a local flood map for each
of the six reference return periods. Simulations are based on the local inertia solver of
LISFLOOD-FP developed by Bates et al. (2010), which is now available as open-source software
(https://www.seamlesswave.com/LISFLOOD8.0). We use the CCM DEM as elevation data, the
synthetic hydrographs described in Section 2.2 as hydrological input data, and a mosaic of
CORINE Land Cover for the year 2016 (Copernicus LMS, 2017) and Copernicus GlobCover
(Global Land Cover Map) for the year 2009 (Bontemps et al., 2009) to estimate the friction
coefficient based on land use.

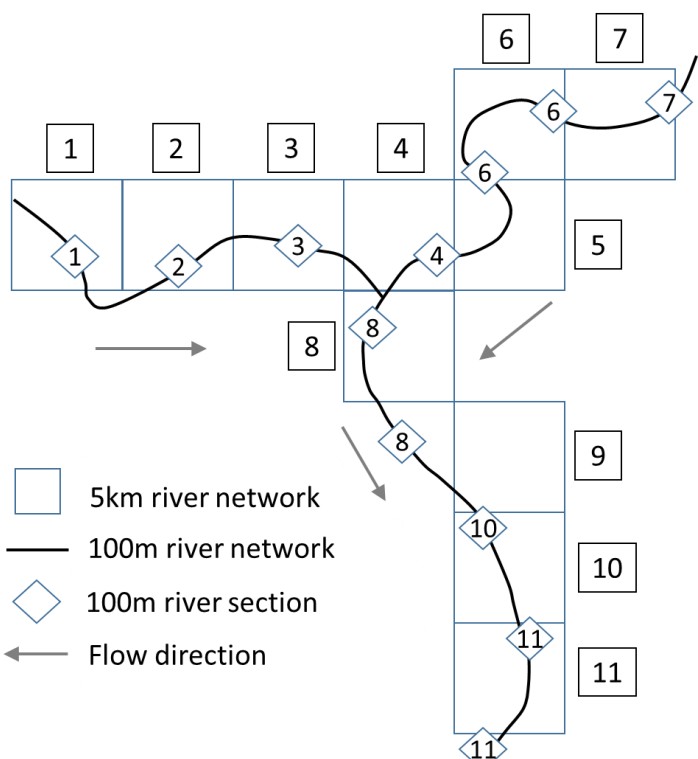


*Figure 3. Conceptual scheme of the EFAS river network (5 km, squares) with the high-resolution*
*network (100 metres) and river sections (diamonds) where flood simulations are derived. The*
*related sections of the two networks are indicated by the same number. Source: Dottori et al.*
*(2017).*

Finally, the flood maps with the same return period are merged together to obtain the continental-
scale flood hazard maps. The 100 metres river network is included as a separate map in the dataset,
to delineate those water courses that were considered in creating the flood hazard maps.
It is important to note that the flood maps developed do not account for the influence of local
flood defences, in particular dyke systems. Such limitation has been dictated mainly by the
absence of consistent data at European scale. None of the available DEMs for Europe has the
required accuracy and resolution to embed artificial embankments into elevation data.
Furthermore, there are no publicly available continental or national datasets describing the
location and characteristics (e.g. dyke height, distance from river channel) for flood protections.
Currently available datasets are based on the design return period of flood protection, e.g. the
maximum return period of flood events that protections can withstand before being overrun,
(Jongman et al., 2014; Scussolini et al., 2016). Most of the protection standards reported by these
datasets for Europe are based on empirical regressions derived using proxy variables (e.g. GDP,
land use), with few data based on actual design standards. While these datasets have been applied
to calculate flood risk scenarios (Alfieri et al., 2015) and flood impacts (Dottori et al., 2017), they
have important limitations when used for mapping flood extent. Wing et al. (2017) linked the
flood return period of protection standards with flood frequency analysis to adjust the bank height
of the river channels, however with impaired performance of the model. Moreover, recent studies
for United States suggest that empirical regressions based on gross domestic product and land use
may not be reliable (Wing et al., 2019).
Despite these limitations, maps not accounting for physical flood defences may be applied to
estimate the flood hazard in case of failure of the protection structures, and for flood events
exceeding protection levels.
*2.3 Validation of flood hazard maps*
*2.3.1 Selection of validation areas and maps*
The validation of large-scale flood hazard maps requires the use of benchmarks with one or more
datasets with extension and accuracy commensurate to the modelled maps. For instance Wing et
al. (2017) used the official hazard maps developed for the conterminous United States to evaluate
the performance of two flood hazard models, respectively designed to produce global- and
continental-scale flood maps (see Section 1). In Europe, all EU Member States as well as the UK
have developed national datasets of flood hazard maps for a range of flood probabilities (usually
expressed with the flood return period), following the guidelines of the EU Floods Directive (EC
2007). These maps are usually derived using multiple hydrodynamic models of varying
complexity (AdB Po, 2012) based on high-resolution topographic and hydrological datasets, such
as DEMs of at least 5 metres resolution in England (Sampson et al., 2015), LIDAR elevation data
in Spain (MITECO 2011), and river sections based on LIDAR surveys in the Po River basin (AdB
Po, 2012). Although official maps might be either prone to errors or incomplete (Wing et al 2017),
these are likely to provide higher accuracy than the modelled maps presented here, and therefore
they have been selected as reference maps for the validation. While official flood maps are
generally available online for consultation on Web-GIS services, only a few countries and river
basin authorities make the maps available for download in a format that allows comparison with
geospatial data. Table 1 presents the list of flood hazard maps that could be retrieved and used for
the validation exercise, while their geographical distribution is shown in Figure 1. Note that the
relevant links to access these maps are provided in the Data Availability section.
While more of such official maps are likely to become available in the near future, the maps
considered here offer an acceptable overview of the different climatic zones and floodplain
characteristics of the European continent. Conversely, we could not retrieve national or regional
flood hazard maps outside Europe, meaning the skill of the modelled maps could not be tested in
the arid regions in Northern Africa and Eastern Mediterranean. In Norway, Spain, the UK and the
Po River Basin the official maps take flood defences into account, which are not represented in
the modelling framework. Official maps for England also include areas prone to coastal flooding
events (such as tidal and storm surges). None of the official maps include areas prone to pluvial
flooding, which are therefore not considered in this analysis.
As mentioned in Section 2.3, the modelled maps do not include the effect of flood protections.
Wherever possible, for the comparison exercise we selected either reference flood maps that do
not account for protections (e.g. Hungary) or maps for flood return periods exceeding local
protection standards, assuming that the resulting flood extent is relatively unaffected by flood
defences. For example, the main stem of the Po river is protected against 1-in-200-year flood
events (Wing et al., 2019), whereas protection standards in England and Norway are usually
above 20 years (Scussolini et al., 2016). Reference maps where the extent and design level of
protection are not known (e.g. Spain) have been also included in the comparison to increase the
number of validation areas.

| Country | Geographical extent | Return periods used | Defences included |
|---------|---------------------|---------------------|-------------------|
| Hungary | Country scale | 30 - 100 – 1,000 years | No |
| Italy | Po River Basin | 500 years | Yes |
| Norway | Country scale | 100 years | Yes |
| Spain | Country scale | 10 - 100 - 500 years | Yes |
| UK | England | 100 – 1,000 years | Yes |

*Table 1. Characteristics of the flood hazard maps used in the validation exercise. The links for*
*downloading the maps are provided in the Data Availability section.*

### 285  2.3.2 Performance metrics and validation procedure

The national flood hazard maps listed in Table 1 are provided as polygons of flood extent, with
no information on water depth or on original resolution of data. According to Sampson et al.
(2015), the official flood hazard maps for England are constructed using DEMs of at least 5 metres
resolution, therefore flood extent maps should be of comparable resolution. Reference flood maps
for the Po basin and Spain are likely to have a similar resolution since they are based on LIDAR
elevation data (AdB Po, 2012; MITECO 2011). For the comparison, official reference maps have
been converted to raster format with the same resolution as the modelled maps (i.e. 100 metres),
while the latter have been converted to binary flood extent maps. To improve the comparison
between modelled and reference maps we applied a number of corrections. Firstly, we used the
CORINE Land Cover map to exclude permanent water bodies (river beds of large rivers or
estuaries, lakes, reservoirs, coastal lagoons) from the comparison. Secondly, we restricted the
comparison area around modelled maps to exclude the elements of river network (e.g. minor
tributaries) included in the reference maps but not in the modelled maps. We used a different
buffer extent according to each study area, considering the floodplain morphology and the
variable extent and density of mapped river network. For example, in Hungary we applied a 10-
km buffer around modelled maps to include the large flooded areas reported in reference maps
and to avoid overfitting. In England, we used a 5 km buffer due to the high density of the river
network mapped in the official maps. The buffer is also applied to mask out coastal areas far from
rivers estuaries, because official maps include flood-prone areas due to 1-in-200-year coastal
flood events. We calculated that flood-prone areas inside the 5 km buffer correspond to 73% of
the total extent for the 1-in-100-year flood. For the Po river basin, we excluded from the
comparison the areas belonging to the Adige river basin and the lowland drainage network, which
are not included in the official hazard maps. In Spain and Norway official flood hazard maps have
only been produced where relevant assets are at risk, according to available documentation
[MITECO 2011; NVE 2020]. We therefore restricted the comparison only to areas where official
flood hazard maps have been produced. Table 2 provide the list of parameters used to determine
the areas used for the comparison.

| Test area | Buffer value (reference maps) | Buffer value (modelled maps) |
|---|---|---|
| Hungary | NA | 10 km |
| Po River Basin | NA | See main text |
| Norway | 5 km | 5 km |
| Spain | 5 km | 5 km |
| England | NA | 5 km |

*Table 2. List of parameters used to determine the extent of areas used for comparing reference*
*and modelled maps (NA: buffer not applied).*

We evaluate the performance of simulated flood maps against reference maps using a number of
indices proposed in literature (Bates and De Roo, 2000; Alfieri et al., 2014; Dottori et al., 2016b;
Wing et al., 2017). The hit ratio (HR) evaluates the agreement of simulated maps with
observations and it is defined as:
$$HR = (Fm \cap Fo)/(Fo) \times 100 \qquad\qquad (1)$$
where $Fm \cap Fo$ is the area correctly predicted as flooded by the model, and $Fo$ indicates the total
observed flooded area. HR scores range from 0 to 1, with a score of 1 indicating that all wet cells
in the benchmark data are wet in the model data. The formulation of the HR does not penalize
over-prediction, which can be instead quantified using the false alarm ratio FAR:
$$FAR = (Fm/Fo)/(Fm) \times 100 \qquad\qquad (2)$$
where $Fm/Fo$ is the area wrongly predicted as flooded by the model. FAR scores range from 0
(no false alarms) to 1 (all false alarms). Finally, a more comprehensive measure of the agreement
between simulations and observations is given by the critical success index (CSI), defined as:
$$CSI = (Fm \cap Fo)/(Fm \cup Fo) \times 100 \qquad\qquad (3)$$
where $Fm \cup Fo$ is the union of observed and simulated flooded areas. CSI scores range from 0
(no match between model and benchmark) to 1 (perfect match between benchmark and model).

## 2.4 Additional tests

To choose the best possible methodologies and datasets to construct the flood hazard maps, we
performed a number of tests using recent input datasets, as well as alternative strategies to account
for vegetation effects on elevation data.

### 2.4.1 Elevation data

It is well recognized that the quality of flood hazard maps strongly depend on the accuracy of
elevation data used for modelling (Yamazaki et al., 2017). This is especially crucial for
continental-scale maps, since the quality of available elevation datasets is rarely commensurate
to the accuracy required for modelling flood processes [Wing et al., 2017]. Moreover, high-
resolution and accurate elevation data such as LIDAR-based DEMs cannot be used for reasons of
consistency, since these data are only available for few areas and countries.
The recent release of new global elevation models have the potential to improve the accuracy of
large-scale flood simulations, and hence the quality of flood hazard maps. Here, we test the use
of the MERIT DEM (Yamazaki et al., 2017) within the proposed modelling approach and we
compare the results with those obtained with CCM DEM. The MERIT DEM is based on the
SRTM data, similarly to CCM DEM, but has been extensively corrected and improved through
comparisons with other large-scale datasets, to eliminate error bias, improve data accuracy at high
latitudes (areas above 60° are not covered by SRTM), and compensate for factors like vegetation
cover. Note that areas above 60° in CCM DEM were derived from national datasets, and therefore
these areas are where the two datasets are likely to differ most.

### *2.4.2 Correction of elevation data with land use*

The CCM DEM elevation dataset is mostly based on SRTM data, and so the elevation values can
be spuriously increased by the effect of vegetation canopy in densely vegetated areas, and by
buildings in urban areas. Recent research work has proposed advanced techniques to remove
surface artefacts, based on artificial neural networks (Wendi et al., 2016, Kulp and Strauss, 2018)
or other machine learning methods (Liu et al., 2018; Meadows and Wilson, 2021). Most
approaches correct DEM elevation with higher-accuracy datasets, using auxiliary data such as
tree density and height for correcting vegetation bias (as done for the MERIT-DEM by Yamazaki
et al., 2017), whereas elevation bias in urban areas can be corrected using night light, population
density, or OpenStreetMap elevation data (Liu et al., 2018). Given that improving elevation data
is not the main scope of this work, we opted for applying a simpler method for quickly correcting
the CCM DEM elevation data. Specifically, we use the land cover map derived from CORINE
Land Cover and Copernicus GlobCover to identify densely vegetated areas and urban areas, and
we applied a correction factor as a function of local land use to reduce elevation locally. The
correction factor varies from 8 metres for densely forested areas, to 2 metres for urban areas. Note
that these values are based on the findings of previous literature studies such as Baugh et al.
(2013) and Dottori et al. (2016b), while a formal calibration was not undertaken.

## *3) Results and discussion*

We present the outcomes of the validation exercise by first describing the general results at
country and regional scale (Section 3.1). Then, we discuss the outcomes for England, Hungary
and Spain (Section 3.2), while the Norway and Po river basin case studies are presented in the
Appendix C. We also complement the analysis with additional validation over major river basins
in England and Spain. In Section 3.3 we compare our results with the validation exercise carried
out by Wing et al. (2017) and with the findings of other literature studies. Finally, in Section 3.4
and 3.5 (and Appendix B) we compare the performance of the present and previous versions of
the flood hazard map dataset, and we discuss the results of the tests with different elevation data
and strategies to account for vegetation.

## 3.1 Validation of modelled maps at national and regional scale

Table 3 presents the validation results for each testing area and return period. The performance
metrics are calculated using the total extent of the reference and modelled maps with the same
return period. The first visible outcome is the low scores for the comparisons with reference maps
with high probability of flooding, i.e. low flood return periods ($< 30$ years). Performances improve
markedly with the increase of return periods due to the decrease of false alarm rate (FAR), while
the hit rate (HR) does not vary significantly. In particular, critical success index (CSI) values
approach 0.5 for the low probability flood maps, i.e., for return periods equal or above 500 years.
Considering that most of the reference flood maps include the effect of flood defences (unlike the
modelled maps), these results suggest that the majority of rivers in the study areas may be
protected for flood return periods of around 100 years or less, as indeed reported by available
flood defence databases (Scussolini et al., 2016). Differences between simulated and reference
hydrological input are likely to influence the skill of modelled flood maps and may depend on
several factors such as the hydrological model performance for peak flows, the extreme value
analysis (distribution used for extreme value fitting, length of available time-series) and the design
hydrograph estimation. In the following Sections, we evaluate the modelled hydrological regime
considering the skill of the LISFLOOD long-term simulation and the uncertainty of the extreme
value analysis (see Appendix B2). However, further analysis is difficult as we have no specific
information on the hydrological input used for the reference flood maps (e.g. peak flows,
statistical modelling of extremes, hydrograph shape). High-probability floods are also sensitive
to the method used to reproduce river channels, and the simplified approach used in this study
might underestimate the conveyance capacity of channels (see Section 3.2.2 for an example).
Finally, the better performance for low-probability floods may also depend on floodplain
morphology, where valley sides create a morphological limit to flood extent.




*Table 3. Results of the validation against official flood hazard maps: value of the performance indices at country and regional scale. RP=Return Period, HR=Hit Ratio, FAR= False Alarm Ratio, CSI=Critical Success Index.*

| REGION | RP (years) | HR | FAR | CSI |
|---|---|---|---|---|
| Spain | 10 | 0.58 | 0.65 | 0.28 |
| Hungary | 30 | 0.77 | 0.88 | 0.11 |
| Spain | 100 | 0.63 | 0.44 | 0.42 |
| Hungary | 100 | 0.76 | 0.74 | 0.24 |
| Norway | 100 | 0.70 | 0.72 | 0.25 |
| England | 100 | 0.53 | 0.31 | 0.43 |
| Po River Basin | 500 | 0.60 | 0.13 | 0.56 |
| Spain | 500 | 0.61 | 0.36 | 0.45 |
| Hungary | 1000 | 0.76 | 0.45 | 0.47 |
| England | 1000 | 0.52 | 0.12 | 0.48 |

## 3.2 Discussion of results at national and regional scale

The results in Table 3 highlight considerable differences in the skill of the flood maps across countries and regions. While some differences may arise from the variability of floodplain morphology and model input data, others are attributable to the different methods applied to produce the reference maps (MITECO 2011; NVE 2020). In the following sections we examine in more detail the outcomes for each study area.

### 3.2.1 England

According to Table 3, modelled flood maps tend to underestimate flood extent in England, as visible by the HR values around 0.5 (e.g. out of every two flooded cells, only one is correctly identified as flooded by the model). Such result is confirmed when focusing the analysis on the major river basins of England, as reported in Table 4. Notably, HR has generally marginal or no increases with the increase of return period considered, while FAR values have a marked decrease. Results of reported by Arnal et al. (2019) and summarized in Figure B1 suggest a fair hydrological skill of the LISFLOOD calibration in England, with KGE values generally above 0.5. The difference between estimated and reference discharge annual maxima is also acceptable,

generally below 25%. However, there is not a clear correlation between hydrological and flood
map skill, with some basins (e.g. Thames) showing high KGE values but relatively low CSI
values.
For the Thames basin, the low CSI value is likely influenced by the tidal flooding component
from London eastwards. According to Sampson et al. (2015), the official flood hazard map
assumes a 1 in 200 year coastal flood along with failure of the Thames tidal barrier, whereas our
river flood simulations use the mean sea level as boundary condition and do include storm surge
and tidal flooding. Concurrent fluvial-tidal flooding processes occur in other river estuaries, so
this might reduce the skill of the modelled maps. Furthermore, the Thames catchment is heavily
urbanized and has extensive flood defence and alleviation schemes compared to the other
catchments (Sampson et al., 2015). Both aspects might increase the elevation bias of CCM DEM
and complicate the correct simulation of extreme flood events.

*Table 4. Validation indices in England and in major river basins.*

| Catchments | 100-year RP | | | 1,000-year RP | | |
|---|---|---|---|---|---|---|
| | **HR** | **FAR** | **CSI** | **HR** | **FAR** | **CSI** |
| **England** | **0.53** | **0.31** | **0.43** | **0.52** | **0.12** | **0.48** |
| Ouse | 0.57 | 0.39 | 0.42 | 0.56 | 0.19 | 0.49 |
| Severn | 0.64 | 0.24 | 0.53 | 0.63 | 0.20 | 0.54 |
| Thames, above Lea | 0.56 | 0.46 | 0.38 | 0.55 | 0.23 | 0.47 |
| Trent | 0.63 | 0.28 | 0.50 | 0.59 | 0.06 | 0.57 |
| Tyne | 0.51 | 0.43 | 0.37 | 0.52 | 0.28 | 0.43 |


Besides these results, the visual inspection of reference maps suggest that the underestimation is
partly caused by the high density of mapped river network in the reference maps, in respect to
modelled maps. Indeed, the modelling framework excludes river basins with an upstream basin
area below 500 km$^2$, meaning that EFAS maps only cover main river stems but miss out several
smaller tributaries. This is clearly visible over the Severn and in the upper Thames basins (Figure
4), and might also explain the lower skill in the lowlands of Ouse and Trent rivers, where the
contributions of main river stems and tributaries to the flood extent are difficult to separate.
Including minor tributaries in the flood maps would require either to increase the resolution of
the climatological forcing to reproduce intense local rainfall, or to add a pluvial flooding
component as done by Wing et al. (2017). Finally, areas prone to storm surge and tidal flooding
around river estuaries might further reduce the overall skill of modelled maps, despite the 5 km
buffer applied.

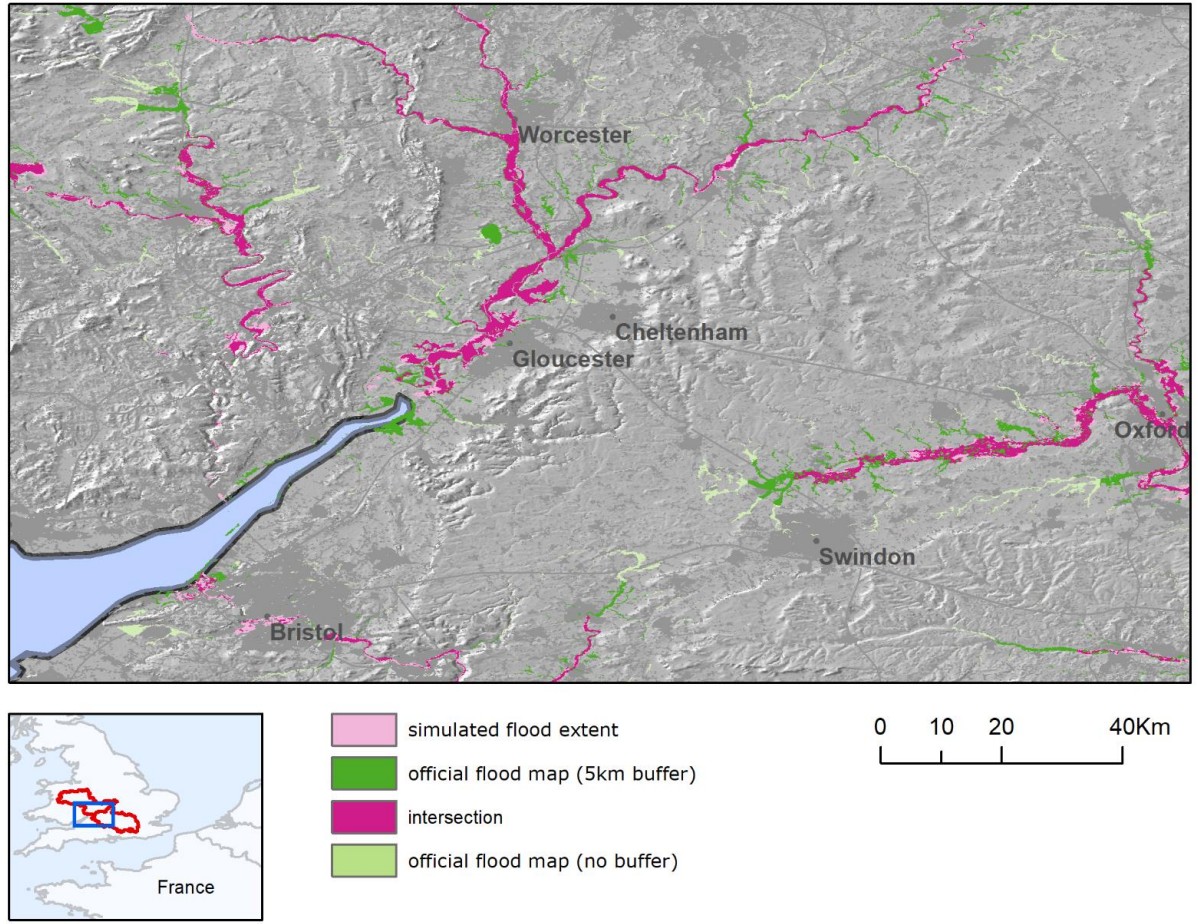


*Figure 4. Comparison of modelled and reference flood hazard maps (1-in-100-year) over the*
*Severn (centre) and the upper Thames (right) river basins in England. Purple areas denotes*
*intersection (agreement) between the modelled and reference set of maps. The original*
*reference maps (i.e. with no masking around modelled maps) are shown in light green.*

*3.2.2 Hungary*
The results in Table 3 for Hungary show a general tendency to overestimate flood extent for all
return periods. HR values are consistently high and do not change much with the return period.
Conversely, FAR is very high for the 1-in-30 year flood map and still considerable even for the
1-in-1000 year flood map. Arnal et al (2019) reported a fair hydrological skill of LISFLOOD
(KGE values >0.5) for the calibration period, even though KGE validation values were
considerably low for the Tisza River. The uncertainty on estimated discharge annual maxima is
also comparable to the average values reported in Appendix B2.
Given that flood defences are not modelled in reference maps, the observed results may be
explained by assuming a large conveyance capacity of river channels. For instance, the 1-in-100
year reference map shows relatively few flooded areas for the Danube main stem (Figure 5), thus
suggesting that the main channels can convey the 1-in-100-year discharge without overflowing.

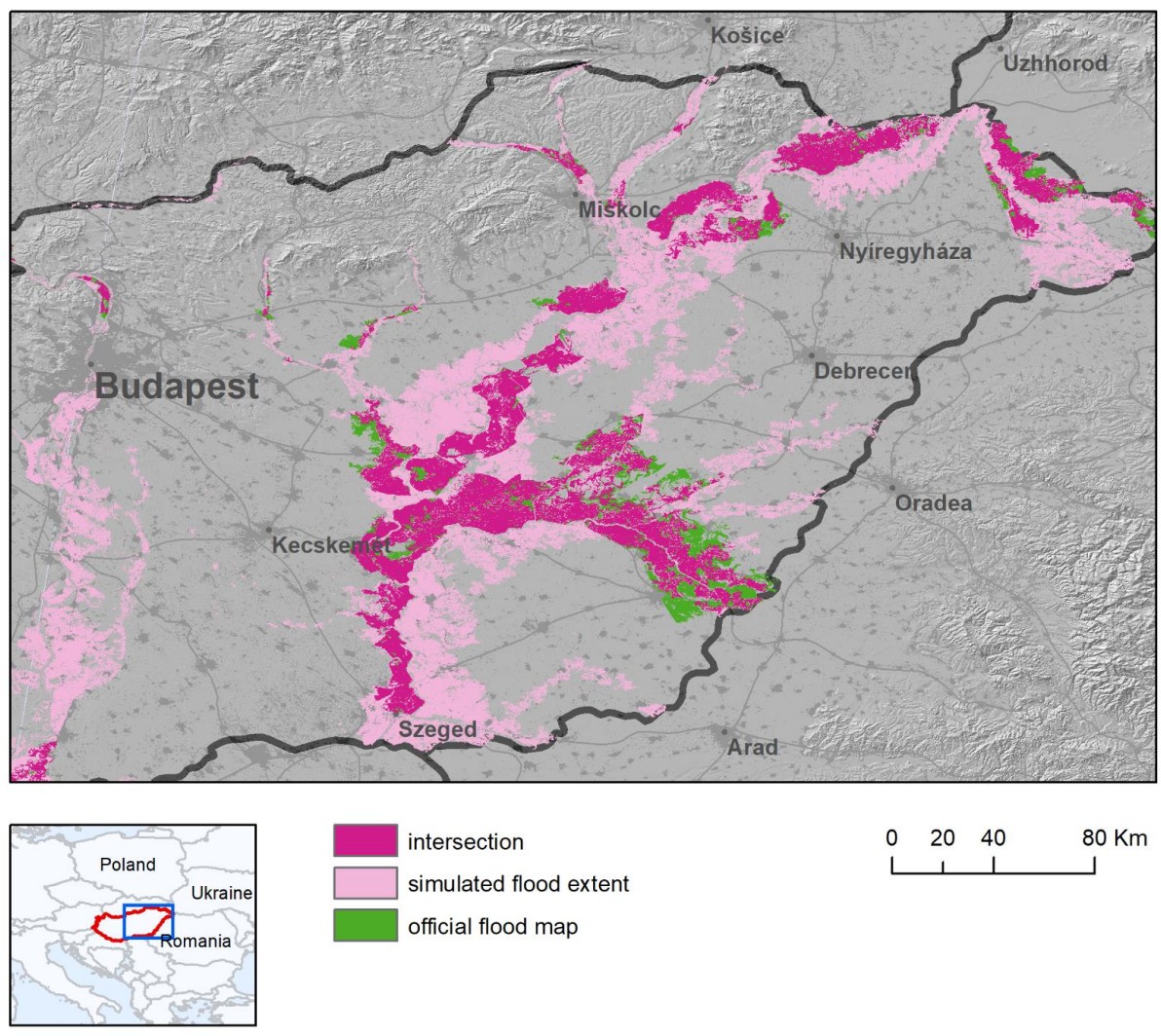


*Figure 5. Comparison of modelled and reference flood hazard maps (1-in-100-year) over the*
*Danube (left) and Tisza (right) rivers in Hungary. Purple areas denotes the intersection between*
*the modelled and reference set of maps.*
Conversely, river channels in the modelling framework are assumed to convey only the 1-in-2-
year discharge. Obviously, the same considerations can be made for 1-in-30-year discharge for
the majority of river network, which explains the very low scores. Furthermore, artificial
structures such as road embankments and drainage network may further reduce flood extent in
lowland areas, leading to further overestimation given the fact that these features are not
represented in the DEM. These findings highlight the need for high-resolution DEM fed with
local-scale information to achieve adequate performance in lowland areas, as observed also by
Wing et al (2019b).
*3.2.3 Spain*
The performance of the modelled maps in Spain show a fairly stable HR value and decreasing
FAR values with increasing return periods, similarly to what was observed for England and
Hungary. The analysis of the results for the major river basins of the Iberian Peninsula, reported
in Table 5, provide further insight on the skill of flood maps. A number of basins exhibit both
large HR and FAR such as the Duero, Tajo and Guadalquivir basins. Rivers in South-East Spain
(Segura, Jucar) have relatively low HR values, while the modelled maps perform better in the
Ebro river basin. The interpretation of results requires the consideration of different aspects.  The
poor results for the 1-in-10-year maps are likely due to the effect of flood protection structures,
such as dykes and flood regulation systems, which are probably relevant also for the 1-in-100-
year map. Indeed, most Iberian rivers are regulated by multiple reservoirs, which are often used
to reduce flood peaks according to specific operating rules. While dykes are not represented in
the inundation model, reservoirs are included in the LISFLOOD model through a simplified
approach, given that operating rules are not known. Therefore, the real and modelled hydrological
regimes might differ significantly, including flow peaks of low-probability flood events. This is
also reflected by the low hydrological skill of LISFLOOD, with KGE values generally below 0.5
with few exceptions (Figure B1).
In addition, the comparison of modelled and reference maps is affected by the partial coverage of
the reference inundation maps in several river basins. According to the information available in
the official website (MITECO 2011) large sections of the river network in the basins of the Duero,
Tajo, Guadiana and Guadalquivir rivers have not been analysed, due to the absence of relevant
assets or inhabited places at risk. Even though this has been accounted for by restricting the area
of comparison around reference maps, a visual inspection of the maps being compared shows
spurious overestimation around the edges of reference map polygons (Figure 6). Finally, the low
HR values scored in rivers in South-East Spain (Segura, Jucar) are partially explained by the
presence of several tributaries not included in EFAS maps.

*Table 5. Validation indices in Spain and in some test river basins.*

| Catchments | 10-year RP | | | 100-year RP | | | 500-year RP | | |
|---|---|---|---|---|---|---|---|---|---|
| | **HR** | **FAR** | **CSI** | **HR** | **FAR** | **CSI** | **HR** | **FAR** | **CSI** |
| **Spain** | **0.58** | **0.65** | **0.28** | **0.63** | **0.44** | **0.42** | **0.61** | **0.36** | **0.45** |
| Duero | 0.60 | 0.74 | 0.22 | 0.65 | 0.55 | 0.36 | 0.65 | 0.46 | 0.42 |
| Ebro | 0.71 | 0.46 | 0.45 | 0.75 | 0.27 | 0.59 | 0.74 | 0.23 | 0.61 |
| Guadalquivir | 0.67 | 0.66 | 0.29 | 0.69 | 0.49 | 0.42 | 0.66 | 0.46 | 0.42 |
| Guadiana | 0.52 | 0.63 | 0.28 | 0.60 | 0.42 | 0.42 | 0.61 | 0.31 | 0.48 |
| Jucar | 0.32 | 0.89 | 0.09 | 0.53 | 0.46 | 0.36 | 0.51 | 0.39 | 0.39 |
| Tajo | 0.60 | 0.85 | 0.14 | 0.70 | 0.63 | 0.32 | 0.69 | 0.49 | 0.41 |
| Segura | 0.18 | 0.89 | 0.07 | 0.38 | 0.52 | 0.27 | 0.41 | 0.24 | 0.36 |

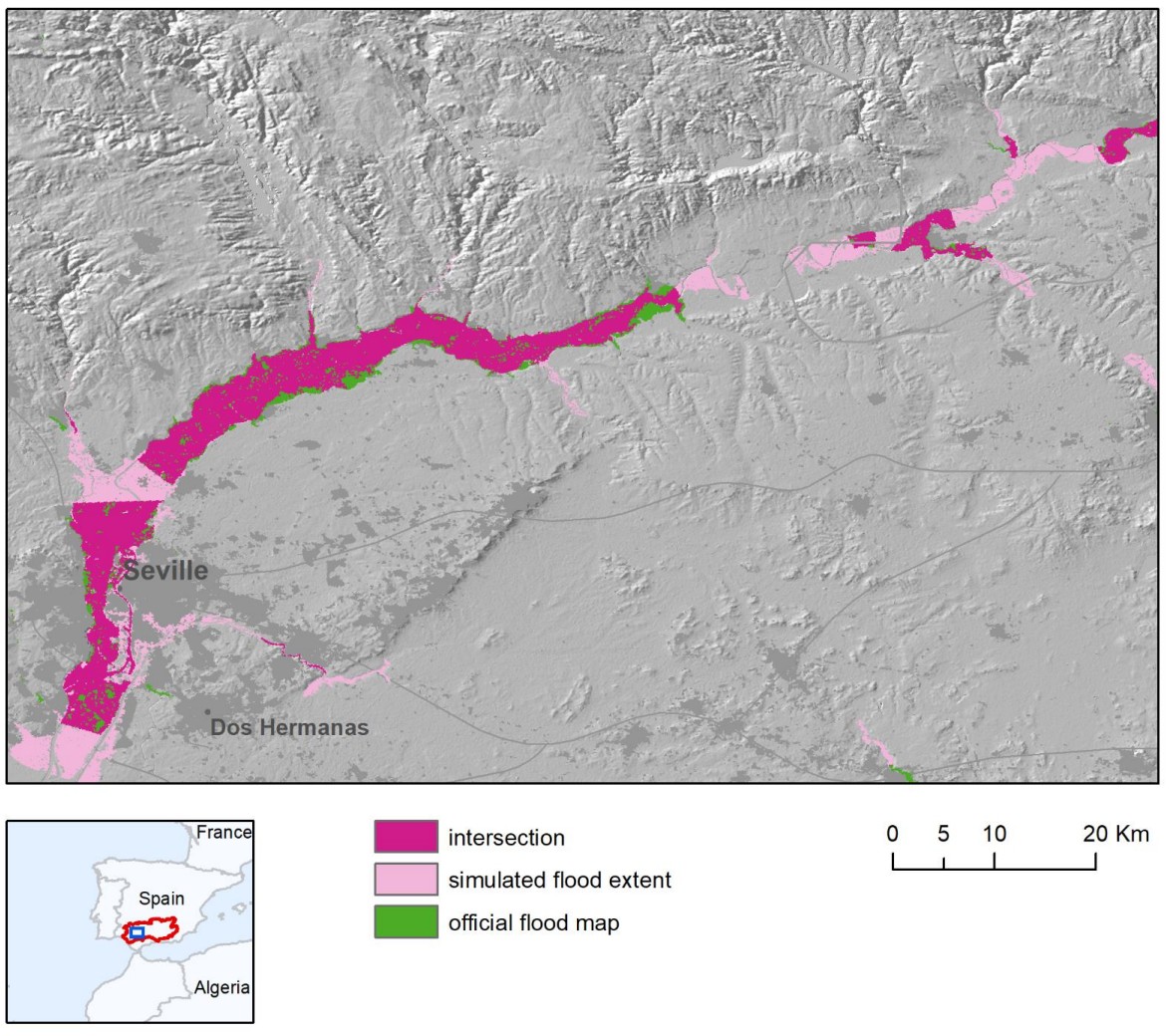


*Figure 6. Comparison of modelled and reference flood hazard maps (1-in-100-year) over a stretch of the Guadalquivir river basin, Spain. Purple areas denote the intersection between the two set of maps.*

### 3.3 Comparison with previous continental-scale validation studies

To put the previously described results in context, we compare them with the validation exercises performed by Sampson et al. (2015) over the Thames and Severn rivers in England, and by Wing et al. (2017) over the United States. The study by Wing et al. is, to our knowledge, the first study that carried out a consistent validation of modelled flood hazard maps at the continental scale. Bates et al. (2021) have recently updated the work by Wing et al. by including pluvial and coastal flooding components in the modelling framework, but their work is not considered here. A

comparison of validation metrics of the three studies are shown in Table 6 and 7. For our
framework, we calculated each index in Table 6 using the overall modelled and reference flood
extent available for each return period (e.g. the value for the 100-year maps includes reference
and modelled maps for England, Spain and Norway). As such, each area is weighted according
to the extent of the corresponding flood map.
As can be seen in Table 6, the continental-scale model by Wing et al. achieved the highest scores
both for 100-year and 500-year return periods. However, this model is based on national datasets
with higher accuracy and resolution than those available for the European continent (e.g. a 10
metres resolution DEM and a detailed catalogue of flood defences). The global and European
models have comparable hit rates for the 100-year flood maps (0.68 and 0.65 respectively), but
the former exhibits a much lower FAR value (0.34 compared to 0.61 for the European model),
and a higher HR value for the 500-year maps.

*Table 6. Comparison of the performance metrics for the European model described in the present*
*study and the two models evaluated in the study by Wing et al. (2017).*

| | RP (years) | HR | FAR | CSI |
|---|---|---|---|---|
| US model (Wing et al.) | 100 | 0.82 | 0.37 | 0.55 |
| Global model (Wing et al.) | 100 | 0.69 | 0.34 | 0.50 |
| European model (this study) | 100 | 0.66 | 0.61 | 0.32 |
| US model (Wing et al.) | 500 | 0.86 | na | na |
| Global model (Wing et al.) | 500 | 0.74 | na | na |
| European model (this study) | 500 | 0.61 | 0.24 | 0.51 |
| European model (this study) | 1000 | 0.68 | 0.39 | 0.47 |


The higher HR values scored by the US and global models might depend on the higher density of
the modelled river network, which includes river reaches up to 50 km$^2$ by simulating both pluvial
and fluvial flooding processes. The lower FAR values of the US and global models might be
explained by the inclusion of flood defences. In the US model, defences are explicitly modelled
using the US dataset of flood defences, while the global model parameterizes flood defences
through the adjustment of channel conveyance using socioeconomic factors and degree of
urbanization (Wing et al., 2017). However, Wing et al observed that the latter methodology had
a negligible effect on HR values in defended areas, when compared with an undefended version
of the model.
Another possible reason for the low FAR values is the different approach used in the validation
method. Wing et al. applied a narrow 1 km buffer around official maps to constrain the area of
comparison and avoid spurious over-prediction in areas not considered by official maps.
However, this might result in a reduction of true false alarms, because part of overestimated flood
areas can go undetected. To verify this hypothesis, we recalculated the performance indices
against the 100-year reference map in Spain using a 1 km buffer instead of the 5 km previously
applied to constrain the validation area. As a result the FAR dropped from 0.44 to 0.34, similar
to the performance of the global model. However, we observed a reduction of true false alarms,
especially in river basins with continuous map coverage such as the Ebro, Jucar and Segura.
The comparison of HR, FAR and CSI values show better scores for the global maps by Sampson
et al. (2015) in respect to our modelled maps (Table 7).

*Table 7. Comparison of the performance metrics for the maps described in the present study and*
*the global maps by Sampson et al. (2015). Metrics for the latter study are calculated removing*
*all channels with upstream areas of less than 500 km2.*

|  | HR | FAR | CSI |
|---|---|---|---|
| Thames (this study) | 0.56 | 0.46 | 0.38 |
| Thames (Sampson et al. 2015) | 0.73 | 0.3 | 0.56 |
| Severn (this study) | 0.64 | 0.24 | 0.53 |
| Severn (Sampson et al. 2015) | 0.83 | 0.23 | 0.67 |


The different masking applied to reference flood maps may explain some of the differences:
Sampson et al. removed all channels with upstream areas of less than 500 $km^2$, whereas here we
use a simpler 5 km buffer around modelled maps. The exclusion of permanent river channels in
our comparison may further penalize the overall score especially for the Thames, which as a rather
large channel estuary. Besides these differences in the validation, the better metrics of the maps
by Sampson et al. may depend on a more accurate hydrological input (based on regionalization
of gauge station data) and a better correction of urban elevation bias (based on a moving window
filter instead of the constant correction values applied here).
To provide further context, the US model by Wing et al. (2017) attained average CSI values of
~0.75 against a number of detailed local models, whereas flood models built and calibrated for
local applications may achieve CSI scores up to 0.9 when benchmarked against very high quality
data (see Wing et al., 2019a). Fleischmann et al. (2019) recently proposed that regional-scale
models can provide locally relevant estimates of flood extent when CSI > 0.65. Although the
overall values shown in Table 3 are consistently below this threshold, better results are observed
for a number of river basins, as shown in Tables 4 and 5.

### 578 *3.4 Comparison with the previous flood map dataset*

Table 8 compares the performances of the flood hazard maps described in the present study
(version 2) with the previous version developed by Dottori et al. (2016a; version 1). The
comparison is shown for England and Hungary. Results for all other areas are comprised within
the range of results shown in Table 3. As can be seen, differences are generally reduced across
the different areas and return periods. Version 1 of the flood maps produced slightly better results
in Hungary for the 100- and 1000-year return period (increased CSI and HR, lower FAR), while
version 2 has somewhat improved performances in England, mainly driven by higher HR.

*Table 8. Comparison of performances of the flood hazard maps described in the present study*
*and developed by Dottori et al. (2016a). Table reports the ratio between flood extents (F2/F1)*
*and the difference between Version 2 and 1 of the HR, FAR and CSI values.*

|  | RP (years) | F2/F1 | ΔHR | ΔFAR | ΔCSI |
|---|---|---|---|---|---|
| Hungary | 30 | 0.97 | -0.5% | -0.4% | 2.9% |
| Hungary | 100 | 1.00 | -2.1% | 0.7% | -2.4% |
| Hungary | 1000 | 1.01 | -3.6% | 5.7% | -6.3% |
| England | 100 | 1.05 | 9.4% | 1.7% | 7.3% |
| England | 1000 | 1.04 | 8.2% | -1.1% | 7.7% |


These outcomes may be interpreted considering the changes in input data between the two
versions, and the structure of the modelling approach and of input data, which in turn has not
changed substantially. The main difference between the two map versions is given by the
hydrological input, with Version 2 using the latest calibrated version of the LISFLOOD model.
For the 100-year return period, peak flow values of Version 2 are on average 35% lower than
Version 1 in Hungary, and 16% lower in England. However, similar decreases are also observed
for the 1-in-2-year peak discharge that determines full-bank discharge. The resulting reduction in
channel hydraulic conveyance in respect to Version 1 is likely to offset the decrease of peak flood
volumes, which explain the small difference in overall flood extent given by the F2/F1 parameter
in Table 8. Such results confirm the low sensitivity of the modelling framework to the
hydrological input observed by Dottori et al. (2016) and by Trigg et al (2016) for a global-scale
application. This low sensitivity is likely to offset the uncertainty related to the estimation of peak
flow values reported in Appendix B. The results also confirm that the knowledge of river channel
geometry is crucial to correctly model the actual channel conveyance and thus improve inundation
modelling. Other differences in input data are given by minor changes in Manning's parameters
and in the EFAS river network, which might contribute to the observed differences.
## *3.5 Influence of elevation data*
Table 9 compares the metrics calculated with CCM DEM elevation data against the same metrics
for the modelled flood maps based on MERIT-DEM. The comparison is carried out for England,
Hungary and the Po river basin. Performance is slightly improved by the use of MERIT-DEM
data for all areas and return periods, in particular through the reduction of FAR, even though the
overall increase of CSI values is limited to few percentage points.

*Table 9. Comparison of performances of the flood hazard maps described in the present study*
*and developed by Dottori et al. (2016a) based on the MERIT-DEM (a) and CCM-DEM (b). Table*
*reports the ratio between flood extents F and the differences for HR, FAR, and CSI (e.g. (HRa-*
*HRb)/HRa ).*

|  | RP (years) | ΔF | ΔHR | ΔFAR | ΔCSI |
|---|---|---|---|---|---|
| Hungary | 100 | -5.3% | 0.0% | -2.0% | 5.1% |
| Hungary | 1000 | -5.9% | -0.1% | -7.6% | 5.2% |
| England | 100 | 0.0% | 2.6% | -5.7% | 3.8% |
| England | 1000 | 1.7% | 2.8% | -7.8% | 3.2% |
| Po | 500 | 0.2% | 0.9% | -4.3% | 3.4% |


Because of this limited improvement and the considerable amount of time required to re-run the
complete set of flood hazard maps (several days for each return period) it was decided not to
update the flood maps using the MERIT-DEM as elevation data. Moreover, new high-resolution
datasets such as the Copernicus DEM (ESA-Airbus 2019), the 90m version of TanDEM-X dataset
(https://geoservice.dlr.de/web/dataguide/tdm90),  and MERIT-HYDRO (Yamazaki et al., 2019)
have recently become available, and therefore future research could focus on performing
additional comparisons to identify which dataset is most suitable for inundation modelling in
Europe.

## 628    *4) Conclusions and ongoing work*

We presented here a new dataset of flood hazard maps covering the geographical Europe and
including large parts of the Middle East and river basins entering the Mediterranean Sea. This
dataset significantly expands the previous available flood maps datasets at continental scale
(Alfieri et  al., 2014; Dottori et al., 2016a), and therefore constitutes a valuable source of
information for future research studies and flood management, especially for countries where no
official flood hazard maps are available. The new maps also benefit from updated models and
new calibration and meteorological data. The maps are being used for a range of applications at
continental scale, from evaluating present and future river flood risk scenarios, to the cost-benefit
assessment of different adaptation strategies to reduce flood impacts, and for comparisons
between different regions, countries and river basins (Dottori et al, 2020b). Moreover, the flood
hazard maps are designed to be integrated with the Copernicus European Flood Awareness
System (EFAS), and will be used to perform operational flood impact forecasting in EFAS
(Dottori et al., 2017).
We performed a detailed validation of the modelled flood maps in several European countries
against official flood hazard maps. The resulting validation exercise is the most complete
undertaken so far for Europe to our best knowledge, and provided a comprehensive overview of
the strengths and limitations of the new maps. Nevertheless, the unavailability of reference flood
maps outside Europe did not allow any validation in the arid regions in North Africa and Eastern
Mediterranean. In these areas, further research will be needed to better understand the
performance of the flood mapping procedure here proposed. Modelled maps generally achieve
low scores for high and medium probability of flooding. For the 1-in-100-year return period, the
modelled maps can identify on average two-thirds of reference flood extent, however they also
largely overestimate flood-prone areas in many regions, thus hampering the overall performance.
Performances improves markedly with the increase of return period, mostly due to the decrease
of the false alarm rates. In particular, critical success index (CSI) values approach and in some
cases exceed 0.5 for return periods equal or above 500 years, meaning that the maps can correctly
identify more than half of flooded areas in the main river stems and tributaries of different river
basins.
It is important to note that the validation was affected by problems in identifying the correct areas
for a fair comparison, because of the different density of the mapped river network in reference
and modelled maps. In our study we used large buffers to constrain comparison areas, which
possibly penalized the model performance by generating spurious false alarms in areas not
considered by official maps. However, we observed that the proposed maps achieve comparable
results to other large-scale flood models when using similar parameters for the validation.
The low skill of modelled maps for high and medium probability of flooding, with large
overestimations observed in different lowland areas, is likely motivated by the non-inclusion of
flood defences in the modelling framework and the simplified representation of channel hydraulic
conveyance, due to the absence of datasets at European scale describing river channels and
defence structures (i.e. design standards and location of dyke systems). Such information
combined with high-resolution DEM fed with local-scale information (artificial and defence
structures) is crucial to improve the performance of large-scale flood models and apply more
realistic flood modelling tools, as observed also by Wing et al (2017, 2019b). Uncertainty in peak
flow estimation can also influence the skill of the modelled maps; however, we found that the
limited sensitivity of the modelling approach to changes in the hydrological input smooths out
this uncertainty source, because channel conveyance is linked to streamflow characteristics. Such
finding highlight the need for independent data of river channel width, shape and depth to better
reproduce streamflow and flooding processes. Moreover, the improved results offered by the use
of the MERIT-DEM elevation data suggest that recent high-resolution datasets such as the
Copernicus     DEM     (ESA-Airbus     2019),     TanDEM-X
(https://geoservice.dlr.de/web/dataguide/tdm90), and MERIT-HYDRO (Yamazaki et al., 2019)
may offer a viable solution to improve future versions of continental-scale flood hazard maps in
Europe.
Increasing map coverage by including the minor river network is likely to improve the skill of
modelled maps. However, this might require the use of a different modelling approach to account
for pluvial flooding (Wing et al., 2017; Bates et al., 2021), along with reliable model climatology
to represent small-scale precipitation processes. Improving the simulation of reservoirs may also
reduce the difference between the real and modelled hydrological regimes in regions such as the
Iberian Peninsula and the Alps.
*Data availability*
The dataset described in this manuscript is accessible as part of the data collection "Flood Hazard
Maps    at    European    and    Global    Scale"    at    the    JRC    Data    Catalogue
(https://data.jrc.ec.europa.eu/collection/id-0054).
Please refer to the dataset as follows: Dottori F., Bianchi A., Alfieri, L., Skoien, J., Salamon P.:
River flood hazard maps for Europe and the Mediterranean Basin. JRC Data Catalogue,
http://data.europa.eu/89h/1d128b6c-a4ee-4858-9e34-6210707f3c81 , doi: 10.2905/1D128B6C-
A4EE-4858-9E34-6210707F3C81, 2020.
The dataset comprises the following maps (eight in total), each one available as a raster (Geotiff)
file:
• Map of permanent water bodies for Europe and the Mediterranean Basin
• River network in Europe and the Mediterranean Basin
• River flood hazard maps for Europe and the Mediterranean Basin (return periods of 10, 20,
50, 100, 200 and 500 years)
The official flood hazard maps used for the validation exercise are freely accessible at the
following web-sites:
• Spain: https://www.miteco.gob.es/es/cartografia-y-sig/ide/descargas/agua/zi-lamina.aspx (in
Spanish);
• Po River Basin: https://pianoalluvioni.adbpo.it/mappe-del-rischio-2/download-mappe/ (in
Italian);
• Norway: https://www.nve.no/flaum-og-skred/kartlegging/flaum/ (in Norwegian);
• England: https://data.gov.uk/dataset/bed63fc1-dd26-4685-b143-2941088923b3/flood-map-
for-planning-rivers-and-sea-flood-zone-3 ; https://data.gov.uk/dataset/cf494c44-05cd-4060-
a029-35937970c9c6/flood-map-for-planning-rivers-and-sea-flood-zone-2 (in English)
• Hungary: https://www.vizugy.hu/index.php?module=content&programelemid=62 (in
Hungarian)
The LISFLOOD hydrological model used in this research is released as open-source software and
available at https://ec-jrc.github.io/lisflood/ (last access on 14 February 2022).
The streamflow dataset derived from the long-term run of the LISFLOOD model is available at
https://cds.climate.copernicus.eu/cdsapp#!/dataset/efas-historical .
The LISFLOOD-FP hydrodynamic model used in this research is available as open-source
software at https://www.seamlesswave.com/LISFLOOD8.0 for research and non-commercial
purposes.
All links have been accessed on 14 February 2022.

## Appendix A: Meteorological observations used for LISFLOOD simulations

The long-term run of the hydrological model LISFLOOD is based on observed data from
meteorological stations and precipitation datasets, which are collected and continuously expanded
as part of the development work for EFAS. The meteorological variables considered are:
precipitation, minimum and maximum temperature, wind speed, solar radiation and vapour
pressure. The number of stations with available meteorological observations depends on the
period and variable considered, with an increasing availability towards the end of the historical
simulation period. As an example, for the year 2016 the number of daily observations available
ranged from ~8.800 for temperature to ~5.500 for precipitation and ~3.700 for vapour pressure.
The input from meteorological stations is completed by a number of precipitation datasets
(EURO4M-APG, INCA-Analysis Austria, ERA-Interim GPCP corrected and Carpat-Clim; for
details see Arnal et al., 2019). Note that the same datasets are used to drive the LISFLOOD
calibration and to calculate the initial conditions for the EFAS forecasts. The data from
meteorological stations and gridded datasets were then interpolated using the interpolation
scheme SPHEREMAP to produce meteorological grids with a daily time step. The reader is
referred to Arnal et al. (2019) for further details.

## Appendix B: Calibration and validation of hydrological components

### B1: LISFLOOD calibration and validation results

We report here an overview of the results of the LISFLOOD calibration and validation presented by Arnal et al. (2019). The skill of LISFLOOD in reproducing observed flow regimes (hydrological skill) is expressed using two indices, the Kling-Gupta Efficiency (KGE; Gupta et al., 2009) and the Nash-Sutcliffe Efficiency (NSE; Nash and Sutcliffe, 1970). The NSE index is widely applied in literature and is useful to measure the hydrological skill under high-flow conditions, given its sensitivity to flow extremes (Krause et al., 2005). The KGE index provides a more complete evaluation of the model skill under variable flow conditions, and is therefore useful for calibration purposes (Gupta et al., 2009; Knoben et al., 2019)

Table B1 summarizes the results of KGE and NSE indices, and Figure B1 shows the spatial distribution of the KGE index values across the EFAS domain. The spatial distribution of NSE is roughly similar. For a detailed list of scores for all stations, please refer to Arnal et al. (2019).

*Table B1. Overview of the hydrological skill of LISFLOOD for the calibration and validation stations.*

| NSE | calibration | | validation | | KGE | calibration | | validation | |
|---|---|---|---|---|---|---|---|---|---|
| | no. of stations | [%] | no. of stations | [%] | | no. of stations | [%] | no. of stations | [%] |
| **> 0.75** | 147 | 21% | 101 | 14% | **> 0.75** | 303 | 42% | 174 | 24% |
| **> 0.5–0.75** | 277 | 39% | 207 | 30% | > 0.5**–0.75** | 240 | 33% | 235 | 33% |
| **> 0.2–0.5** | 165 | 23% | 171 | 25% | **> 0.2–0.5** | 91 | 13% | 172 | 24% |
| **> 0–0.2** | 35 | 5% | 65 | 9% | > 0**–0.2** | 36 | 5% | 44 | 6% |
| **≤0** | 93 | 13% | 153 | 22% | **≤0** | 47 | 7% | 73 | 10% |
| | ∑ **717** | | ∑ **698** | | | ∑ **717** | | ∑ **698** | |

As can be seen from Table B1, 75 % of all stations scored a KGE higher than 0.5 during calibration, and 57 % during validation. NSE index values above 0.5 are scored for 60% and 44% of stations, respectively for the calibration and validation periods.

It is clearly noticeable that the skill is not homogeneously distributed across Europe, with higher
skills in large parts of Central Europe, and lower skill mostly in Spain caused by the strong
influence of reservoirs and flow control structures. The other study areas considered in the
validation exercise (England, Hungary, Norway, Po river basin) exhibit KGE and NSE values
generally above 0.5.


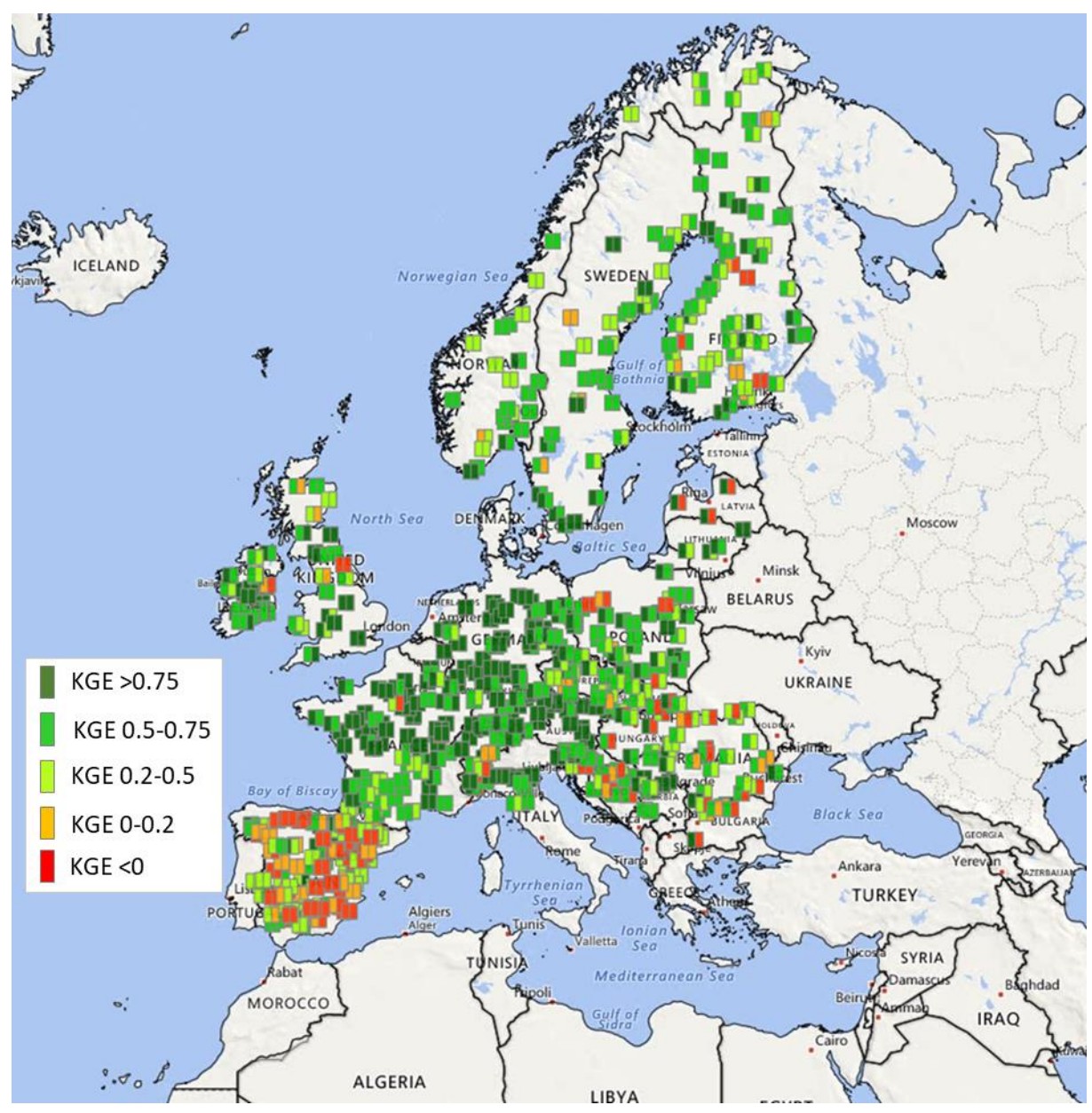

*Figure B1. Hydrological skill of EFAS at the calibration locations. Colour coding denotes the*
*quality of the KGE during calibration (left half of square) and validation (right half of the square).*
*Adapted from Arnal et al. (2019).*



## B2: performance of the extreme value analysis

Here we evaluate the performance of the Gumbel distribution in fitting the available reference discharge values (26 annual maxima calculated for all the grid points of the LISFLOOD long-term run). Specifically, we compare the empirical and fitted distributions of streamflow annual maxima using the Cramer-von Mises test (Anderson, 1962), and we calculate the average differences between reference and fitted discharge values. Table B2 summarizes the resulting p-values over the study area. Figure B2 compares empirical and fitted distributions in two locations of the rivers Rhine and Danube.

*Table B2. Overview of the performance of the Gumbel distribution calculated with the Cramer-Vin Mises criterion.*

| P value | % LISFLOOD points |
|---|---|
| <0.1 | 5% |
| 0.1–0.25 | 6% |
| 0.25–0.5 | 14% |
| 0.5–0.75 | 23% |
| >0.75 | 52% |

*Figure B2. Comparison of the empirical and fitted distributions of annual discharge maxima at selected locations of the rivers Rhine (left) and Danube (right).*

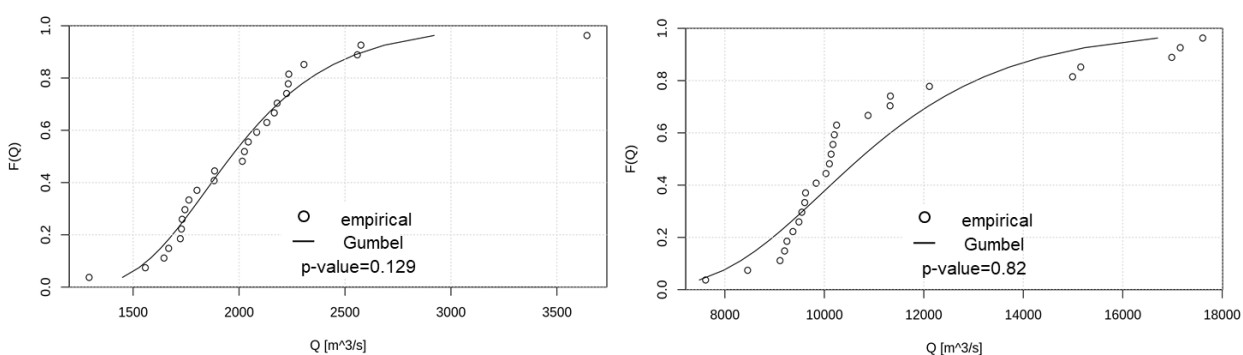

P-values in Table B2 suggest a low skill of the fitted Gumbel distributions; however the resulting uncertainty in the estimates of discharge maxima is generally below 25%, as in the examples

shown in Figure B2. This is considered acceptable because the reference discharge maxima are
modelled and not observed values. Due to limited sample size, it is not possible to evaluate the
extrapolation error for peak flows beyond the available sample; however, previous studies
suggested the suitability of the Gumbel distribution. Cunnane (1989) stated that the Gumbel
distribution is effective for small sample sizes, whereas the Generalized Extreme Value (GEV)
distribution shows a better overall performance if the size is greater than 50. More recently,
Papalexiou and Koutsoyiannis (2013) found similar results for extreme precipitation values. In
particular, they demonstrated that short record lengths affects the estimation the GEV shape
parameter, and thus the choice between a two-parameter (Gumbel) and a three-parameter GEV.
Di Baldassarre et al. (2009) observed that the Gumbel distribution might estimate flood extremes
with high return periods (e.g. 100-year) with smaller errors than other distributions, if the
available sample size is small. Further research could use longer observed streamflow series to
compare different extreme value distributions across European regions, similarly to what done by
Villarini and Smith (2010) for the eastern United States and Rahman et al. (2013) for Australia.

*Appendix C: Additional results*
*C1: validation of the hazard maps for the Po River Basin*
According to Table 3, the modelled flood maps provide a better reproduction of reference maps
for the Po River, compared to other study areas. False alarms are low, while hit ratio (HR) values
indicate that two out of every three pixels in the reference map are correctly identified as flooded.
The analysis of reference and modelled maps (Figure C1), suggests that the underestimation is
partly caused by flooded areas along some tributaries which are not included in modelled maps.
Other areas with omission errors are located near confluences of the Po main stem and the major
tributaries in Emilia-Romagna, which may depend on the underestimation of peak flow on
tributaries. In fact, the results of the LISFLOOD calibration in Figure B1 show better hydrological
skill along the Po main stem, compared to some tributaries. Finally, it is likely that the inclusion
of smaller tributaries of the river network in the modelled maps would improve the overall
performance.

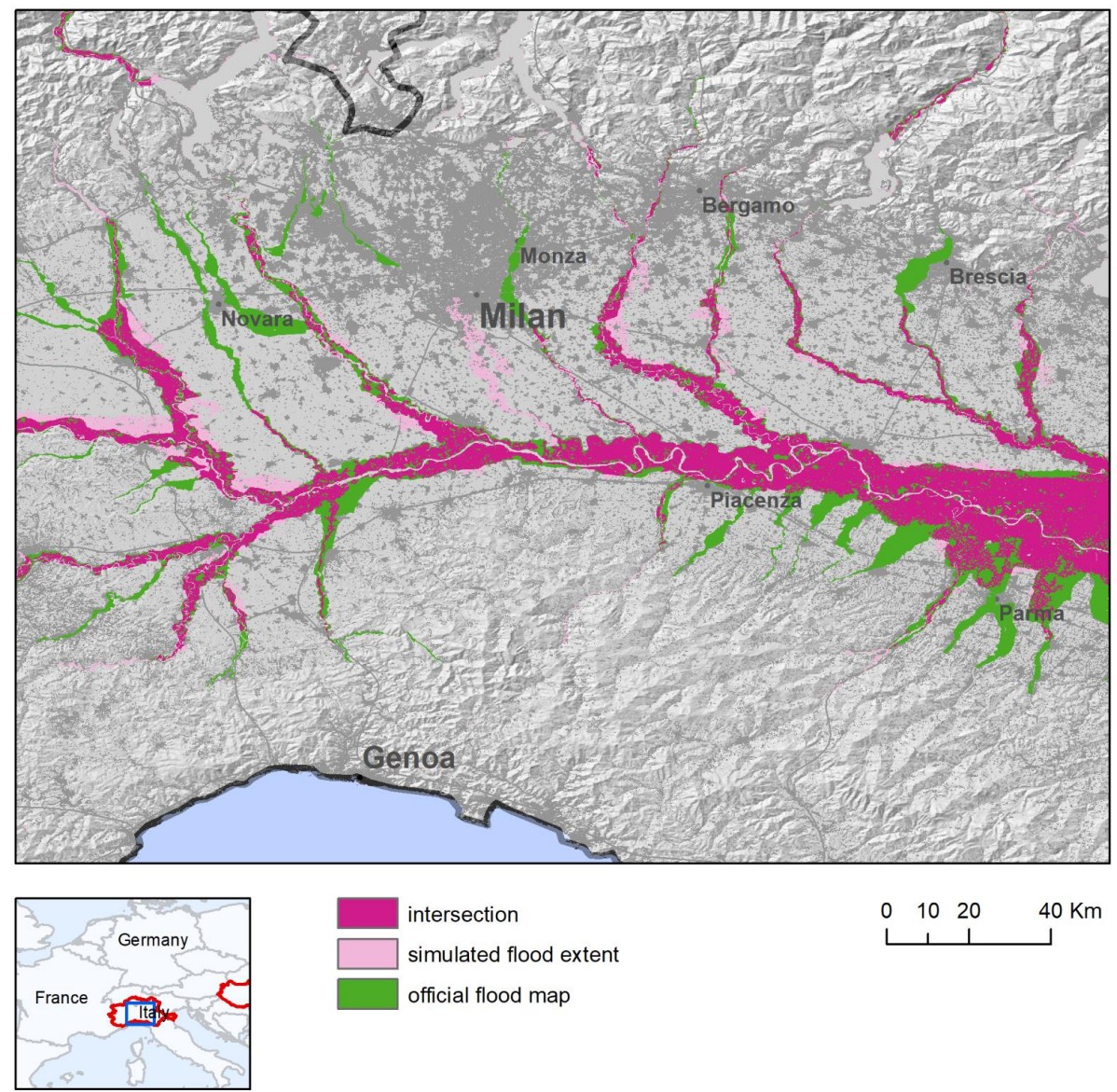


*Figure C1. Comparison of modelled (blue) and reference (green) flood hazard maps (1-in-500-*

*year) over the Po river basin, Italy. Purple areas denotes the intersection (agreement) between*
*the two set of maps.*

## C2: validation of the hazard maps for Norway

The results of the modelled flood maps in Norway show a general tendency to overestimate flood
extent for the 1-in-100-year events, with high values for both hit ratio (HR) and false alarm ratio
(FAR). Such a result is in fact largely influenced by the relatively small extent and discontinuous
coverage of reference maps. Flood-prone areas for the 1-in-100-year official maps only cover 215
km$^2$, possibly due to the low density of populated places in Norway, while they cover between
4700 and 5700 km$^2$ for England, Spain and Hungary. As for Spain, we applied a 5km buffer to
restrict the area of comparison around reference maps, yet this leads to spurious overestimation
around the edges of reference map polygons. Notably, the performance improves markedly with
the use of a 1km buffer as in Wing et al., (2017), which results in increased critical success index
(CSI) scores up to nearly 0.50.
The results of reported by Arnal et al. (2019) and summarized in Figure B1 suggest an acceptable
hydrological skill of the LISFLOOD calibration in Norway, with a majority of gauge stations
scoring KGE values above 0.5. In the areas with lower scores, the model performance for low-
probability flood events might be influenced by an incorrect estimation of peak discharges driven
by snow melt, which plays a relevant role in determining low-probability flood events.

### *C3: Influence of correcting elevation data with land use*

We tested the results of correcting CCM DEM elevation data with vegetation cover in
Scandinavia, where the percentage of land covered by forests is more relevant than in the other
regions included in the modelled flood maps. For the 1-in100-year flood maps, the overall
difference in flood extent between the corrected and uncorrected maps is less than 4%, and similar
values were found for the other return periods. Moreover, the HR, FAR and CSI values   of two
set of maps differ by less that 2% when calculated against the 1-in100-year official map in
Norway, probably because forested areas have not been considered as relevant flood-prone areas.
These results suggest that the simulation of densely vegetated areas have a limited importance in
determining the overall performance of modelled flood maps in Europe.

## *Author contribution*


FD: conceptualization, formal analysis, investigation, data curation writing (original draft, review
and editing); LA: methodology, investigation, writing (review and editing); AB: data curation,
validation, visualization; JS: investigation, writing (review and editing); PS: conceptualization,
project administration, writing (original draft, review and editing)

## *Competing interests*

The authors declare that they have no conflict of interest.


## *Acknowledgements*

This study has been partially funded by the COPERNICUS programme and by an administrative arrangement with Directorate General 'European Civil Protection and Humanitarian Aid Operations (DG ECHO) of the European Commission. EFAS is operated and financed as part of the Copernicus Emergency Management Service. The authors would like to thank Niall McCormick for his valuable suggestions on the early versions of the manuscript

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
