# Peer review of "A new dataset of river flood hazard maps for Europe and the Mediterranean Basin"

_Earth System Science Data, 2020_

## Author Response (AR1)

**Reply to Reviewers**

**RC1: 'Comment on essd-2020-313', Anonymous Referee #1, 09 Apr 2021**

This paper evaluates new European flood maps provided by the Copernicus European Flood Awareness System. The methods behind the mapping have been developed and published elsewhere and are only briefly documented here (too briefly in a few places – see comments below). The advantage of the presentation approach is that the paper should be accessible to non-experts in this type of modelling.

The paper focuses on the performance of the hazard layers against several national flood hazard maps and international studies using a similar class of regional flood models. The paper is well presented and easy to follow. It makes a useful contribution to the literature and more validation studies of this type are needed. Generally the conclusion are well supported by the analysis, although the early focus on the Mediterranean basin region is lost later in the manuscript. I agree fully with the premise of the paper and would recommend publication subject to the revisions outlined below.

*We thank the Referee for his/her positive comments on our manuscript.*

Line 81: Add reference for LISFLOOD-FP for consistency with LISFLOOD. There is also now source code published for LISFLOOD-FP and you could cite this later for consistency with the presentation of LISFLOOD https://doi.org/10.5194/gmd-2020-340

*We thank the Reviewer for providing this reference, which is now included in the main text as well as in the Data Availability section*

Line 140: I appreciate that the article is trying to avoid repeating technical details published elsewhere. However, I would like a little more detail on the statistical analysis of the extreme flows to be presented. Specifically what data were used (AMAX)? What distributions were fitted? Alfieri et al 2014 describe the overall method but I also thing so direct citation to the extreme value analyse method followed would be useful.

*We applied the statistical analysis of extreme flows over the long-term hydrological simulation generated with LISFLOOD. For each pixel of the river network, we extracted annual maxima for the period 1990-2016 and we used the L-Moments approach to fit a Gumbel distribution and calculate extreme flow return periods. This explanation has been placed at the beginning of the new section 2.2 "Hydrological input of flood simulations".*

Line 146: What is the source of the high-resolution river network data?

*The high-resolution river network data is taken from the CCM River and Catchment Database for Europe, the same database of the DEM (Vogt et al., 2007). We have added this reference at the beginning of Section 2.3.*

Line 157: Does the DEM include building and vegetation or are these removed to approximate a DTM?

*Thanks for raising this point. The elevation values of CCM DEM are influenced by buildings and vegetation cover, as already mentioned in Section 2.4.2. . This motivated our quick correction of elevation values using land use information. In the revised text we have rewritten Section 2.4.2 to clarify the importance of these source of uncertainty on DEM values.*

Line 161: Given that a 2D model is used without the river channels how are river flows accounted for? For example, is a design flow subtracted from the volume entering the model (this approach would approximate the method JBA used for the original 1 in 1000 year extreme hazard map for the UK), is zero in channel flow assumed (this would potentially put more water onto the floodplain than in reality and results in a somewhat precautionary model), are the channels represented as 1D components (for example the approach taken in LFPtools https://doi.org/10.1016/j.envsoft.2019.104561 ), or do you burn a channel into the DEM (tricky at 100 m resolution on smaller rivers and usually only used for high-resolution simulations). I don't believe it matters which approach was taken from a publication perspective, but a short note is needed here to acknowledge and justify the choice made, especially as there are examples of LISFLOOD-FP being applied in all of these ways.

*In each flood simulation we modify the input flow hydrograph by subtracting the 1-in-2-year flood discharge value to all daily time step values, therefore reducing the overall flood volume entering the model. Conversely, the original DEM is not modified, following the approach proposed by Alfieri et al. (2014,2015). We have added this piece of information in the expanded description of input hydrographs (see end of updated Section 2.2).*

Line 243: How are the reference maps treated in the comparison. Do they maintain some native resolution or are the polygons rasterised to the same 100 m resolution as the modelled maps. Assuming 100 m resolution what impact does this have, I assume any loss of resolution usually makes the reference maps easier to fit?

*All the national flood hazard maps used in this work are available as polygons of flood extent, and the original resolution is not known. As such, we could not quantify the impact of coarsening reference flood maps, however it is likely to be negligible compared to the overall error in modelled flood maps. Reference maps were then converted to raster format with the same resolution as the modelled maps, while the latter have been converted to binary flood extent maps (as explained in the original manuscript).*

Line 272: Did you consider using a DEM derived from TanDEM-X data?

*We are aware of the recently released 90m DEM derived from TanDEM-X data (https://geoservice.dlr.de/web/dataguide/tdm90/ ), but it was not considered in this work due to time constraints. We have mentioned the possibility of using this dataset for future research in the Conclusions, along with Copernicus DEM and MERIT-Hydro datasets.*

Line 287: Some acknowledgement that the approach taken to the vegetation and urban correction is far from the state of the art is needed here. Some of the newer machine learning based approaches are likely to do a significantly better job or removing surface

artifacts than this approach and we know that flood simulation is very sensitive to the quality of vegetation and building removal in the global DEM's. I do not think this detracts from the value of the study, but it should be clear that there are known routes to potentially better modelling here.

*We take the point. Indeed, working on the improvement of elevation data was not the main scope of this work (and the MERIT DEM applied in the tests already includes a correction of vegetation effects), so we modified Section 2.4.2 to make clear that we proposed a rather basic and quick approach. We agree on that mentioning more advanced research on this topic would benefit the manuscript. To this end, we have included the following text to Section 2.4.2 :" Recent research works proposed advanced techniques to remove surface artifacts, based on artificial neural networks (Wendi et al., 2016, Kulp and Strauss, 2018) or other machine learning methods (Liu et al., 2018; Meadows and Wilson, 2021). Most approaches compare the DEM to be corrected with higher-accuracy datasets, using auxiliary data such as tree density and height for correcting vegetation bias (Yamazaki et al., 2017), whereas elevation bias in urban areas can be corrected using night light, population density, or Open Street Map elevation data (Liu et al., 2018)".*

Line 309: depending on the approach taken to represent the river channel network, see comment above, this might also be significant for the simulation of higher probability floods. Channels are very important flow pathways and especially so for smaller floods.

*We fully agree with the Reviewer on this point. We have included his/her remark in the revision of Section 3.1 as follows: "Considering that most of the reference flood maps include the effect of flood defences (contrary to the modelled maps), these results suggest that the majority of rivers in the study areas may be protected for flood return periods around 100 years or lower, as indeed reported by available flood defence databases (Scussolini et al., 2016). High-probability floods are also sensitive to the method used to reproduce river channels, and the simplified approach used in this study might underestimate the conveyance capacity of channels (see Section 3.2.2 for an example). Finally, the better performance for low-probability floods may depend on floodplain morphology, where valley sides create a morphological limit to flood extent."*

General: How did you deal with coastal areas in the England flood maps? These are 1 in 200 year return periods in the flood map but also this source is not included in your modelling.

*We used a 5km buffer around modelled flood maps to mask out coastal areas far from rivers estuaries, because the geo datasets available did not allow separating areas prone to coastal or river flooding. Moreover, the estuary area of several rivers show concurrent fluvial-tidal flooding, as mentioned for the Thames by Sampson et al. (see following comments), and this interaction is again not simulated in our modelling framework. We have mentioned the fact that reference maps for England include areas prone to 1-in-200-year coastal flood events (revised Section 2.3.2), and we further clarify this point in the description of results for England (section 3.2.1)*

Line 330: Thames will have significant tidal flooding from London eastwards.

*Thanks for this valuable insight. We used it to improve the comment of results for England (section 3.2.1) as follows: "there's not a clear correlation between hydrological and flood map skill, with some basins (e.g. Thames) showing high KGE values but relatively low CSI values. For the Thames basin, the low CSI value is likely influenced by tidal flooding component from London eastwards. According to Sampson et al. (2015), the official flood hazard map assumes a 1 in 200 year coastal flood along with failure of the Thames tidal barrier, whereas river flood simulations use the mean sea level as boundary condition and do include storm surge and tidal flooding. Given that other river estuaries are characterized by concurrent fluvial-tidal flooding processes, this might reduce the skill of the modelled maps."*

Line 339: The smaller tributaries, and coastal flooding issues is discussed for the Thames and Severn in Sampson et al 2015. I think that would be a better comparison/citation specifically in this section than Wing et al 2017. Their CSI values from the Sampson global flood model might also be useful to report for these basins and compare with your values to complement

Line 345: see comments above.

*We thank the Reviewer for the useful suggestion. We included the considerations by Sampson et al. about the influence of smaller tributaries, coastal flooding and urban areas to expand the discussion of Section 3.2.1. Moreover, we added a comparison of performance metrics between our maps and Sampson et al. in Section 3.3.*

Figure 4: Could you include floodplains outside of the 5 km buffer in another colour? TBH this map doesn't really reflect how much flooding in the UK is not being simulated by this modelling setup – which is absolutely fine but the paper should be upfront about it.

*We have modified Figure 4 to include flood-prone areas outside the 5km buffer. Moreover, we calculated that flood-prone areas inside the 5km buffer correspond to 73% of the total extent for the 1-in-100-year flood map (as specified now in Section 2.3.2 of the revised manuscript).*

The EA flood map doesn't include surface water flooding from pluvial flooding, that would be an even more detailed layer, so the flooding missed is fluvial and coastal.

*We now mention in Section 2.3.1 that areas prone to pluvial flooding are missing from our analysis in all study areas.*

Line 410: this is an unfair comment given the publication date, but there is an updated US validation in Bates et al 2021 WRR. I don't think this would have any significant impact on the discussion here but it might be worth citing.

*We thank the Referee for the suggestion. We have included this recent work in the discussion of Section 3.3.*

I've no experience with the flood maps outside of the UK but the comparisons undertaken look robust.

*We thank the Referee for the appreciation*

The conclusions are well supported by the analysis, however little validation has been undertaken around Mediterranean basins, particularly those areas into which the new maps have extended. Flood simulation in arid areas are often more challenging and the performance from Europe might not translate well to North Africa and the Eastern Mediterranean. I think some discussion of this issue is needed given the focus on the Mediterranean basin region in the title and introduction… Or perhaps less focus on the Mediterranean basin region and more on Europe earlier in the manuscript if the discussion is going to be too vague in this regard.

*In the revised text we now highlight the issue of incomplete validation of the dataset. For instance, in the Abstract we state that "Further verification in North African and Eastern Mediterranean regions is needed to better understand the performance of the flood maps in arid areas outside Europe" while the end of the Introduction we state that "The number and extent of the validation sites allows for a more detailed evaluation in respect to previous efforts by Alfieri et al. (2014) and Paprotny et al. (2017), even though none of the validation sites is located outside Europe due the unavailability of national flood maps". In the conclusions, we say that " (…) the unavailability of reference flood maps outside Europe did not allow any validation in the arid regions in North Africa and Eastern Mediterranean. In these areas, further research will be needed to better understand the performance of the flood mapping procedure here proposed."*

*However, we believe that mentioning the Mediterranean basin region in the title and introduction is justified and constitutes a major point of the work, because the dataset is the first example of high-resolution (100m) and freely available flood hazard maps available for the whole Mediterranean basin region, including North Africa and the Eastern Mediterranean and.*

**Additional References**

*Bates, P. D., Quinn, N., Sampson, C., Smith, A., Wing, O., Sosa, J., et al. (2021). Combined modeling of US fluvial, pluvial, and coastal flood hazard under current and future climates. Water Resources Research, 57, e2020WR028673.* [https://doi.org/10.1029/2020WR028673](https://doi.org/10.1029/2020WR028673)

*Kulp, S.A.; Strauss, B.H. CoastalDEM: A global coastal digital elevation model improved from SRTM using a neural network. Remote Sens. Environ. 2018, 206, 231–239.*

*Liu, Y., Bates, P.D., Neal, J.C. and Yamazaki, D., 2019, December. Bare-earth DEM Generation in Urban Areas Based on a Machine Learning Method. In AGU Fall Meeting Abstracts (Vol. 2019, pp. H41N-1899).*

*Meadows, M.; Wilson, M. A Comparison of Machine Learning Approaches to Improve Free Topography Data for Flood Modelling. Remote Sens. 2021, 13, 275.* [https://doi.org/10.3390/rs13020275](https://doi.org/10.3390/rs13020275)

*Wendi, D.; Liong, S.-Y.; Sun, Y.; Doan, C.D. An innovative approach to improve SRTM DEM using multispectral imagery and artificial neural network. J. Adv. Model. Earth Syst. 2016, 8, 691–702.*

**RC1: 'Comment on essd-2020-313', Anonymous Referee #1, 09 Apr 2021**

The authors present a dataset of flood hazard maps, updated with respect to the ones presented in Alfieri et al. (2014) and Dottori et al. (2016). Both in the abstract and in the introduction, it is not clear what are the improvements and the differences between the new and the old version of the dataset. Later in the manuscript some information are provided but this point is not clearly discussed. Please add the needed details.

*Following the Referee's comment, we have highlighted in the Introduction the main advances of the new dataset in respect of the previous versions (namely, the extended domain, the use of updated hydrological and hydrodynamic models, and new calibration and meteorological input data). Furthermore, we pointed out the relevance of the validation exercise, that allows for a more detailed evaluation in respect to previous efforts in literature*

Many times the authors refer to previous own works. Even if I understand the reason for which some details are not given in the manuscript, some additional information e.g., about the methodology applied to obtain synthetic flood hydrograph or the meteorological data used ad input to the LISFLOOD model, could be useful for the reader.

*We have expanded the description of the data and methods used for this study, following similar comments from both Referees. The revised text features new sections describing the meteorological dataset driving the LISFLOOD model (Appendix A) and the synthetic flood hydrographs used in flood simulations (Section 2.2 in the main text). Moreover, we provide now more details about the extreme value analysis and the validation procedure. More detailed explanations are given in the following replies*

The results discussed throughout the manuscript should be better explained. In particular, it is not clear how performance scores reported in the Tables are obtained for each study areas and it is not clear how the comparison shown in Table 6 has been carried out. For major details, please refer to specific comments.

*The description of methodology applied to obtain the results has been revised and clarified. Please refer to the following point-by-point reply for more details.*

The authors attribute the differences between modelled and reference flood maps to a number of shortcomings of the modelling framework specifically related to the hydrodynamic simulation. No description (magnitude of peak, duration of the hydrograph) is given about the hydrographs used as input to the hydrodynamic LISFLOOD-FP model that are (could be) different from the ones used to obtain reference flood maps. Please, add details on the hydrological inputs used and comment how they impact on the definition of the flood extension.

*We have added a new section (2.2) with a detailed description of the methodology used to derive the flood hydrographs, reporting the following text: "The input hydrographs necessary for the flood simulations are derived from the LISFLOOD streamflow dataset described in Section 2.1. Streamflow data is available for the EFAS river network at 5 km grid spacing for rivers with upstream drainage areas larger than 500 km². For each pixel of the river network we selected annual maxima over the period 1990-2016 and we used the L-moments approach to fit a Gumbel distribution and calculate peak flow values for reference return periods of 10, 20, 50, 100, 200 and 500 years. Note that we also calculated the 30- and 1000-year return periods in limited parts of the model domain to allow validation against official hazard maps, see Section 2.3. Subsequently, we calculate a Flow Duration Curve (FDC) from the long-term simulation. The FDC is obtained by sorting in decreasing order all the daily discharges, thus providing annual maximum values $Q_D$ for any duration i between 1 and 365 days. Annual maximum values are then averaged over the entire period of data, and used to calculate the ratios $\varepsilon_i$ between each average maximum discharge for i - th duration $Q_{D(i)}$ and the average annual peak flow (i.e. $Q_D$ = 1 day). Design flood hydrographs are derived using daily time steps. The peak value is given by the peak discharge for the selected T- year return period $Q_T$, while the other values Qi are derived multplying $Q_T$ by the ratio $\varepsilon$ i . The hydrograph peak $Q_T$ is always placed in the centre of the hydrograph, while the other values Qi are sorted alternatively to produce a triangular hydrograph shape, as shown in Figure xx. Because river channels are usually not represented in the CCM DEM, flood hydrograph values are reduced by subtracting the 2-years discharge peak, which is commonly considered representative of river bank-full conditions. Note that the original DEM is not modified. The total duration of the hydrograph is given by the local value of the time of concentration Tc, therefore all the durations > Tc are discarded from the final hydrograph."*

*Differences between simulated and reference hydrological inputs are likely to influence the skill of modelled flood maps. However, further analyses are difficult because we have no specific information on the hydrological input used for the reference flood maps (e.g. peak flows, hydrograph shape). Along the text, we use the skill of the long-term simulation of LISFLOOD to evaluate the agreement between modelled and observed hydrological regime, but this does not necessarily translate to extreme values (See also the following comments for more details). These considerations have been included at the beginning of Section 3.1.*

Moreover, it is expected that higher differences are found for basins with properties and characteristics not well described by the approximations used in the procedure. For instance, if a leveed river is simulated without considering flood defence structures, the identified maps will be different form those that can be obtained by using a detailed morphology description. Actually, the considered simplifications can modify significantly the flooding dynamics. A comment of the authors is required.

*We fully agree with the Referee on this point. Indeed, the influence of not including defence structures in the simulations is already discussed in the manuscript at different points (see L 175-195, 304-308, 383-385). Also, the limitations given by the simplified representation of*

*river channels are discussed in Section 3.2.2. We have carefully revised the manuscript to make sure that all the limitations of the modelling framework are clearly stated.*

Specific comments:

Line 19 and 164: "six different flood return 20 periods…". Reading Line 141, seems that the analysed return periods are seven. Please, modify the manuscript where needed.

*To clarify this, we have modified the paragraph and moved it in the new section 2.2. The text now reads as follows: "For each pixel of the river network we selected annual maxima over the period 1990-2016 and we used the L-moments approach to fit a Gumbel distribution and calculate peak flow values for reference return periods of 10, 20, 50, 100, 200 and 500 years. Note that we also calculated the 30- and 1000-year return periods in limited parts of the model domain to allow validation against official hazard maps, see Section 2.3." In other words, The 30- and 1000-year return period flood maps were produced only for Hungary and England for comparison exercise while the hazard maps for the other six return periods cover the whole domain.*

Lines 30-31. What does the authors mean with "large variability"?

*The original text in the abstract (" In addition, the large variability of reference maps affects the correct identification of the areas for the validation, thus penalizing scores") will be replaced by the following: " In addition, the different design of reference maps (e.g. extent of areas included) affects the correct identification of the areas for the validation, thus penalizing scores"*

Lines 104-108. Please, add some details on the meteorological observations used to force the LISFLOOD model. Moreover, what does the authors mean with "the static input maps have been updated and expanded"? Please, specify.

We rephrased this part in Section 2.1 as follows: *"The long-term run of LISFLOOD is driven by gridded meteorological maps, derived by interpolated meteorological observations from stations and precipitation datasets (see Appendix A for details). The meteorological dataset has been updated in respect to the dataset used by Dottori et al. (2016a) to include new stations and gridded datasets across the new EFAS domain (Arnal et al. 2019). In addition, LISFLOOD simulations require a number of static input maps such as land cover, digital elevation model, drainage network, soil parameters and parameterization of reservoirs. All the static maps have been updated to cover the whole EFAS domain depicted in Figure 1."*

*Moreover, we have included a new section Appendix A describing the meteorological forcing of the LISFLOOD model, taken from the report by Arnal et al. (2019). "The long-term run of the hydrological model LISFLOOD is based on observed data from meteorological stations and precipitation datasets, which are collected and continuously expanded as part of the development work for EFAS. The meteorological variables considered are: precipitation, minimum and maximum temperature, wind speed, solar radiation and vapour pressure. The number of stations with available meteorological observations depends on the period and variable considered, with an increasing availability towards the end of the historical simulation period. As an example, for the year 2016 the number of daily observations*

*available ranged from ~ 8.800 for temperature to ~ 5.500 for precipitation and ~ 3.700 for vapour pressure. The input from meteorological stations is completed by a number of precipitation datasets (EURO4M-APG, INCA-Analysis Austria, ERA-Interim GPCP corrected and Carpat-Clim; for details see Arnal et al., 2019). Note that the same datasets are also used to drive the LISFLOOD calibration and to calculate the initial conditions for the EFAS forecasts. The data from meteorological stations and gridded datasets were then interpolated using the interpolation scheme SPHEREMAP to produce meteorological grids with a daily time step. The reader is referred to Arnal et al. (2019) for further details."*

Lines 140-145: Please, add details on this part to allow the reader understanding the procedure. How the statistical analysis has been carried out? how the synthetic flood hydrographs have been defined?

*Following previous comments from both Referees, we have added a new section (2.2) providing more details on the statistical analysis of extremes and the derivation of flood hydrographs.*

Lines 146-160: this part should be better explained and modified as it is very similar to paragraph 2.21 in Dottori et al. (2017). Please rephase.

*We rewrote this part (now in Section 2.3) as follows: "The continental-scale flood hazard maps are derived from local flood simulations run all along the river network as in Alfieri et al. (2014). We use the DEM at 100 m resolution developed for the Catchment Characterization and Modelling Database (CCM; Vogt et al., 2007) to derive a high-resolution river network at the same resolution. Along this river network we identify reference sections every 5 km along stream-wise direction, and we link each section to the closest upstream section (pixel) of the EFAS 5km river network, using an partially automated procedure to ensure a correct linkage near confluences. In this way, the hydrological variables necessary to build the flood hydrographs can be transferred from the 5km to the 100m river network. Figure 3 describes how the 5km and 100m river sections are linked using a conceptual scheme.*
*Then, for every 100 m river section we run flood simulations using the 2D hydrodynamic model LISFLOOD-FP (Shaw et sl., 2021), to produce a local flood map for each of the six reference return periods. Simulations are based on the local inertia solver of LISFLOOD-FP developed by Bates et al. (2010), which is now available as open-source software (https://www.seamlesswave.com/LISFLOOD8.0). We use the CCM DEM as elevation data, the synthetic hydrographs described in Section 2.2 as hydrological input, and a mosaic of Corine Land Cover for the year 2016 (Copernicus LMS, 2017) and GlobCover for the year 2009 (Bontemps et al., 2009) to estimate the friction coefficient based on land use.*
*Finally, the flood maps with the same return period are merged together to obtain the continental-scale flood hazard maps. The 100m river network is included as a separate map in the dataset, to delineate which water courses have been considered in the creation of the flood hazard maps."*

Lines 215-216: this sentence seems to be in contrast e.g., with the results shown in Table 3 for Spain for return period of 10 years.

*This was indeed not consistent in the previous version. We modified this paragraph in Section 2.3.1, which now reads as follows: "For the comparison exercise, we selected available maps for return periods for which flood extent is likely to be less conditioned by flood defences. For instance, the main stem of the Po river is protected against the 1-in-200-year flood events (Wing et al., 2019), whereas protection standards in England and Norway are usually above 20 years (Scussolini et al., 2016). Conversely, we consider the 1-in-30-year map for Hungary and the 1-in-10-year map for Spain because flood defences are either not accounted for (Hungary) or their extent and design level is not known (Spain)".*

Line 223: which is the native resolution of the official reference maps? After the conversion to raster format, which is the adopted resolution to make the comparison with simulated flood maps?

*The official reference maps are provided as polygons with no indication of the original resolution. According to Sampson et al. (2015), the official flood hazard maps for England are constructed using DEMs of at least 5 m resolution, therefore the resolution of reference maps should be similar. Reference flood maps for the Po basin and Spain are likely to have a similar resolution since they are based on LIDAR elevation data. All the reference maps have been converted to 100m resolution for the comparison with modelled maps. All these details are now included in the revised version (section 2.3.2).*

Lines 247, 251 and 255: Please, correct the numbers of the equations.

*Thanks for spotting this inconsistency, we amended the numbers of the equations*

Lines 243-256: Please, add the variability range of HR, FAR and CSI and define the perfect score for each of them.

*We updated the description of the indices according to the Referee's suggestion, with the following information. HR ranges from 0 to 1, with a score of 1 indicating that all wet cells in the benchmark data are wet in the model data. FAR scores range from 0 (no false alarms) to 1 (all false alarms). CSI scores range from 0 (no match between model and benchmark) to 1 (perfect match between benchmark and model).*

Lines 301-301: from Table 3 does not seem that "Performances improve markedly with the increasing of return periods, with a general increase in the hit rate HR". Form Table 3, HR is quite constant with the return period. Please rephrase.

*We rephrased these lines in Section 3.1 as follows: "Performances improve with the increasing of return periods due to the decrease of false alarm rate FAR, while the hit rate HR does not vary significantly."*

Lines 304-310 and throughout the manuscript: the authors comment on the performances of simulated flood maps to reproduce the reference maps, stressing that differences could be ascribed to floodplain morphology, presence of flood defence structure etc.. According to me the differences between simulated and reference flood maps could be ascribed also to the hydrological input routed through the river channel. How different is the flood

hydrograph used by the authors with respect to the one used by to build the refence flood maps? Do the authors have any information on this point?

*Thanks for this comment. We agree on that differences between simulated and reference hydrological inputs could explain some of the observed differences between flood maps. As specified in a previous comment, the revised manuscript now includes a specific section about the hydrological input for flood simulations. Unfortunately, we could not find the necessary data to reconstruct the input flood hydrographs of the reference flood maps (e.g. peak flows, hydrograph shape). For official flood maps in Spain and in the Po river basin, only a description of general methods applied is available online. The evaluation of the skill of the LISFLOOD model can provide some hint on the similarity of modelled and observed hydrological regime, but this does not necessarily translate to extreme values. We added these considerations at the beginning of Section 3.1.*

Table 3: How are obtained the performance indices for the study areas? Are these values obtained as average values? Please, add details.

*The performance indices are calculated using the total extent of the reference and modelled maps with the same return period. As such each score is a single value and it is not averaged. The corresponding text has been updated accordingly.*

Line 316: Should be Table 3.

Lines 323, 354, 355 and throughout the manuscript. Please use the abbreviations for hit rate, false alarm rate and critical success index.

Line 415: should be Table 6.

*We amended the manuscript as suggested*

Lines 415 – 400 and Table 6: how the values in Table 6 have been obtained? Do they refer to all the five study areas? Please, specify.

*We have added the following text at the beginning of Section 3.3 to clarify this point "The performance indices in Table 6 were calculated by first summing up all the reference and modelled maps for the same return period, where available. Then we calculated each index using the overall modelled and reference flood extent (e.g. the value for the 100-year maps includes reference and modelled maps for England, Spain and Norway). As such, each area is weighted according to the extent of the corresponding flood map."*

Table 1-3 and Line 364: for Hungary a 30-y return period has been used from the reference maps. How these maps have been compared with the simulated maps? Reading the manuscript seems that only 10, 20, 50, 100, 200, 500 and 1000 return period have been simulated.

*This was on oversight of the previous version, a 30-year return period flood maps was produced only for Hungary for comparison exercise. We modified the text (now in Section 2.2) as follows: "For each pixel of the river network we selected annual maxima over the period 1990-2016 and we used the L-moments approach to fit a Gumbel distribution and*

*calculate peak flow values for reference return periods of 10, 20, 50, 100, 200 and 500 years. Note that we also calculated the 30- and 1000-year return periods in limited parts of the model domain to allow validation against official hazard maps, see Section 2.3. "*

Figure 3 could be removed from the manuscript and the limits of the test areas could be added in Figure 1.

*We have removed Figure 3 and modified Figure 1 as suggested*

Figure 2: Please, modify the number 10 and 11 in the diamond. Specifically, write them in a horizontal line.

*The Figure has been amended as requested*

Table 1: Please, for each country add the link where the reference flood maps can be downloaded.

*We now mention in the revised text and in the caption of Table 1 that the links for downloading the maps are provided in the Data Availability Section.*

---

## Referee Report (RR1)

**Review for manuscript "A new dataset of river flood hazard maps for Europe and the Mediterranean Basin region"**

**Authors:** Francesco Dottori, Lorenzo Alfieri, Alessandra Bianchi, Jon Skoien, Peter Salamon

**Journal:** ESSD

**Summary**

The authors present a new flood hazard map dataset for Europe and the Mediterranean region for different return periods up to 500 years. They evaluate modeled flood extents using existing national flood hazard maps. They show that the new data product over- or underestimates flood extents depending on the region considered and consider the dataset to be valuable in regions where no more detailed national flood hazard maps exist.

**General remarks**

The paper is clearly structured and has generally a good reading flow. The dataset presented improves the spatial coverage of existing flood hazard maps. However, the paper lacks methodological detail not evident to readers unfamiliar with previous work published by the authors. In particular, important information on hydrological model evaluation with respect to high flows and the statistical models (i.e. Gumbel distribution and hydrograph construction procedure) is missing.

**Major points**

1) What kind of new features that the latest version of the LISFLOOD model have that make it more suitable to derive flood hazard maps than previous versions (l. 83-86)?

2) I guess that the official hazard maps were derived using locally-calibrated hydrological models and are therefore considered to be more reliable than maps derived using global models (and are therefore chosen as a reference). This is not evident though and should be mentioned somewhere (l. 89-91)

3) The authors briefly describe how the LISFLOOD model has been calibrated (l. 124-126), however, I could not find any information about how the model was validated and what the outcome of the validation step was. It would be important to provide some validation results with respect to high-flow simulation performance in order to establish trust into the streamflow simulations used for the hazard assessment.

4) The authors estimate 500- and 1000-year floods using a sample consisting of 26 annual maxima only (l. 151). Such extrapolations are extremely dangerous because of large sampling uncertainty. I would therefore limit the analysis to extremes with 100-year return period.

5) The Gumbel distribution is chosen for extreme value analysis. I have my doubts that this 2-parameter distribution is a good fit for the data. Goodness-of-fit testing is required here to show the suitability of the Gumbel distribution to model annual maxima. If it is rejected in many cases, I would rather use the more flexible GEV distribution.

6) I do not fully understand how the design flood estimates were derived (l.156-165). The description of how event duration is included needs more/clearer explanation because the FDC itself is only a CDF of daily flow and does not really say anything about event duration:

How is event duration derived? How is event volume derived? How is the event shape derived?

7) I find it a inconsistent to compare existing flood risk maps which have been derived taking flood protection measures into account to estimated flood risk maps derived ignoring these measures (l.365-369). This seems as if you were comparing apples with pears instead of apples with apples. Wouldn't a flood risk map not considering protection measures depict a wrong (and overestimated) picture of flood risk?

8) Overall, a more nuanced discussion of different uncertainty sources not limited to uncertainties related to the hydraulic modeling step would be required. Such uncertainty sources include hydrological model performance, statistical modeling, design hydrograph estimation, …

9) Some additional language editing would further improve the reading flow.

**Minor points**

- L. 109: when was LISFLOOD last updated?
- L. 118-120: can you please provide the data sources for all these datasets?
- L. 156: what do you mean by 'long-term' simulation?
- Figure B1: color legend is missing.
- Suggestion for slight title adjustment: 'A new dataset of river flood hazard maps for Europe and the Mediterranean Basin' or 'River flood hazard maps for Europe and the Mediterranean Basin: a new dataset derived using LISFLOOD'

---

## Author Response (AR2)

**Reply to the Editor**

Dear authors,

many thanks for submitting the revised version of the manuscript on flood hazard mapping in Europe. The paper has now been seen by two referees. While referee #1 has now further comments, the new referee #3 did raise a number of important points which have to considered. In particular the questions on model validation, the reliability of extrapolations to 500 and 1000 year events and the choice of the Gumble over the GEV distribution are highly relevant since they affect the overall dependability of the data product.

*Please find below a detailed reply to all the points raised by Referee #3.*

**Reply to Referee #3**

General remarks

The paper is clearly structured and has generally a good reading flow. The dataset presented improves the spatial coverage of existing flood hazard maps. However, the paper lacks methodological detail not evident to readers unfamiliar with previous work published by the authors. In particular, important information on hydrological model evaluation with respect to high flows and the statistical models (i.e. Gumbel distribution and hydrograph construction procedure) is missing.

*We thank the Referee for his/her positive view of our work and for the useful comments. Please find below a detailed reply to all the points raised*

1) What kind of new features that the latest version of the LISFLOOD model have that make it more suitable to derive flood hazard maps than previous versions (l. 83-86)?

*The new LISFLOOD version benefits from updated in the model components, in the input dataset and in the calibration routine. Specifically, we expanded the description of the new LISFLOOD version in Section 2.1 as follows (see lines 120-128): "The new version features an improved routine to calculate water infiltration, the possibility of simulating open water evaporation and several minor adjustments that correct previous code inconsistencies (Arnal et al., 2019)". In addition, most of the input datasets (e.g. meteorological data, digital elevation model etc) have been updated, and a new calibration algorithm has been developed and applied with more calibration and validation data". All these updates are likely to improve the estimation of river flow regimes and hence also the estimation of peak flows.*

2) I guess that the official hazard maps were derived using locally-calibrated hydrological models and are therefore considered to be more reliable than maps

derived using global models (and are therefore chosen as a reference). This is not evident though and should be mentioned somewhere (l. 89-91).

*We thank the Reviewer for pointing out this issue. In the revised manuscript, we have modified Section 2.3.1 to motivate the use of official flood hazard maps in the validation (lines 245-255): "In Europe, all member states of the European Union as well as the United Kingdom have developed national datasets of flood hazard maps for a range of flood probabilities (usually expressed with the flood return period), following the guidelines of the EU Floods Directive (EC 2007). These maps are usually derived using multiple hydrodynamic models of varying complexity (AdB Po 2012) based on high-resolution topographic and hydrological datasets, such as DEMs of at least 5 m resolution in England (Sampson et al., 2015), LIDAR elevation data in Spain (MITECO 2011), and river sections based on LIDAR surveys in the Po River basin (AdB Po, 2012). Even though official maps might be prone to errors or be incomplete (Wing et al 2017), they are likely to provide a higher accuracy than the modelled maps presented here, and therefore have been selected as reference maps for the validation."*

3) The authors briefly describe how the LISFLOOD model has been calibrated (l. 124-126), however, I could not find any information about how the model was validated and what the outcome of the validation step was. It would be important to provide some validation results with respect to high-flow simulation performance in order to establish trust into the streamflow simulations used for the hazard assessment.

*In the revised text (Appendix B) we have included a more detailed description of the joint calibration/validation done by Arnal et al. (2019). In particular, we added an overview of the calibration/validation results using the Nash – Sutcliffe efficiency index (NSE), and a new table summarizing the model skill. The NSE index is more suited to evaluate the model accuracy in simulating discharge peaks and complements the Kling-Gupta efficiency index.*

4) The authors estimate 500- and 1000-year floods using a sample consisting of 26 annual maxima only (l. 151). Such extrapolations are extremely dangerous because of large sampling uncertainty. I would therefore limit the analysis to extremes with 100-year return period.

*We agree with the Referee on that using short time series bring substantial uncertainty in extrapolating peak flow extremes. However, the comparison with the previous version of the flood maps described in Section 3.4 shown that much of the uncertainty in peak flow estimation is actually smoothed out by the low sensitivity of flood extent and depth to return period, in particular for return periods above 100 years. Such low sensitivity was observed by Dottori et al. (2016) and by Trigg et al (2016) for a global-scale application of the same flood hazard mapping procedure, and derives from the relatively low accuracy of the available topographic information, in particular the absence of river channels and structures*

*such as river embankments. We have updated Section 3.4 to include this explanation, which is also mentioned in the Conclusions (lines 671-674).*

5) The Gumbel distribution is chosen for extreme value analysis. I have my doubts that this 2-parameter distribution is a good fit for the data. Goodness-of-fit testing is required here to show the suitability of the Gumbel distribution to model annual maxima. If it is rejected in many cases, I would rather use the more flexible GEV distribution.

*In the revised version Section 2.2. we have added some paragraphs to motivate the choice of the Gumbel distribution: "We used the Gumbel distribution to keep a parsimonious parameterization (2 parameters instead of 3 of the generalized extreme value (GEV), log-normal and other distributions) and thus avoid over-parameterization when extracting high return period maps from a relatively short time series. The same distribution was also adopted for the extreme value analysis in previous studies regarding flood frequency and hazard (Alfieri et al., 2014, 2015; Dottori et al., 2016)." In addition, we mentioned the uncertainty arising from the extreme value analysis (distribution used for extreme value fitting, length of time series etc) in the discussion of results in Section 3 (lines 387-395)*

6) I do not fully understand how the design flood estimates were derived (l.156-165). The description of how event duration is included needs more/clearer explanation because the FDC itself is only a CDF of daily flow and does not really say anything about event duration:How is event duration derived? How is event volume derived? How is the event shape derived?

*We have carefully rewritten the last part of Section 2.1 to better explain how we derive synthetic flood hydrographs : "The synthetic flood hydrographs are derived using daily time steps. The peak value of the hydrograph is given by the peak discharge for the selected T-year return period $Q_T$, while the other values $Q_i$ are derived multplying $Q_T$ by the ratio $\varepsilon_i$. The hydrograph peak $Q_T$ is placed in the centre of the hydrograph, while the other values $Q_i$ are sorted alternatively to produce a triangular hydrograph shape, as shown in Figure 2. The total duration of the synthetic hydrograph is given by the local value of the time of concentration $T_c$, therefore all the durations > $T_c$ are discarded from the final hydrograph (Figure 2). Because river channels are usually not represented in continental scale topography, flood hydrograph values are reduced by subtracting the 2-years discharge peak $Q_T(2)$, which is commonly considered representative of river bank-full conditions (note that the original DEM is not modified with this procedure). Hence, the overall volume of the flood hydrograph is given by the sum of all daily flow values with duration < $T_c$."*

7) I find it a inconsistent to compare existing flood risk maps which have been derived taking flood protection measures into account to estimated flood risk maps derived ignoring these measures (l.365-369). This seems as if you were comparing apples

with pears instead of apples with apples. Wouldn't a flood risk map not considering protection measures depict a wrong (and overestimated) picture of flood risk?

*We fully agree on that the modelled flood hazard maps should be compared with reference maps not accounting for flood protections. To better explain our approach we rewrote part of Section 3.1.2 (lines 269-277): "The modelled maps does not include the effect of protections, as mentioned in Section 2.3. Wherever possible, for the comparison exercise we selected either reference flood maps that do not account for protections (e.g. Hungary) or maps for flood return periods exceeding local protection standards, assuming that the resulting flood extent is little conditioned by flood defences. For instance, the main stem of the Po river is protected against the 1-in-200-year flood events (Wing et al., 2019), whereas protection standards in England and Norway are usually above 20 years (Scussolini et al., 2016). Reference maps where the extent and design level of protection is not known (e.g. Spain) have been also included in the comparison to increase the number of validation areas." Note that in this latter case, we explicitly mention in Section 3.2.3 that the influence of protections can condition the outcomes of the comparison.*

8) Overall, a more nuanced discussion of different uncertainty sources not limited to uncertainties related to the hydraulic modeling step would be required. Such uncertainty sources include hydrological model performance, statistical modeling, design hydrograph estimation, …

*In the revised version, Section 3 we now mention different uncertainty sources related to the elaboration of the hydrological input (lines 387-395): "Differences between simulated and reference hydrological input are likely to influence the skill of modelled flood maps and may depend on several factors such as the hydrological model performance for peak flows, extreme value analysis (distribution used for extreme value fitting, length of available time series) and design hydrograph estimation. However, further analyses are difficult because we have no specific information on the hydrological input used for the reference flood maps (e.g. peak flows, statistical modelling of extremes, hydrograph shape). In the following sections, we use the skill of the LISFLOOD long-term simulation to evaluate the agreement between modelled and observed hydrological regime, but this does not necessarily translate to extreme values."*

*As we state in the revised version, a more in-depth discussion of the hydrological uncertainty is hampered by the lack of information about reference flood maps. This means that we cannot quantify the differences between modelled and reference maps on crucial aspects such as the statistical modelling of extremes, input peak flows, the shape of hydrographs etc. However, we do provide an overview of LISFLOOD performance in all the subsections dedicated to the study areas, and where possible we discuss the possible influence of hydrological input on the skill scores (for instance, in Section 3.2.3 we mention that that modelled flow peaks for low-probability flood events are more uncertain).*

9) Some additional language editing would further improve the reading flow.

*The paper has been carefully edited to improve the language.*

Minor points

L. 109: when was LISFLOOD last updated?

*The LISFLOOD version used to run the hydrological simulations and documented in Arnal et al. (2019) was finalized in the second half of 2018. The code of the model version presently available as source code in GitHub (see Data Availability) is basically unchanged besides bug fixing and the possibility to use 6-hourly time step. To avoid confusion, we change all references to the LISFLOOD version from "the latest version" to "updated version" (in respect to the version used by Dottori et al., 2016).*

L. 118-120: can you please provide the data sources for all these datasets?

*We modified the text to state that all the data sets are described in Arnal et al (2019)*

L. 156: what do you mean by 'long-term' simulation?

*We refer here to the LISFLOOD long-term simulation described in Section 2.1. In the revised paper we replaced it with the term "streamflow dataset" and we specify at the beginning of Section 2.1 that it is derived from LISFLOOD long-term simulation.*

Figure B1: color legend is missing.

*The colour legend has been added to the Figure.*

Suggestion for slight title adjustment: 'A new dataset of river flood hazard maps for Europe and the Mediterranean Basin' or 'River flood hazard maps for Europe and the Mediterranean Basin: a new dataset derived using LISFLOOD'

*We thank the Referee for the suggestion and we modified the title to "A new dataset of river flood hazard maps for Europe and the Mediterranean Basin". We prefer not to include the LISFLOOD model in the title, because it is not the only model applied (flood simulations are run using the hydrodynamic model LISFLOOD-FP) and because the overall procedure include several steps not directly related to LISFLOOD.*

**References**

Alfieri, L., Salamon, P., Bianchi, A., Neal, J., Bates, P.D., Feyen, L., 2014. Advances in pan-European flood hazard mapping, Hydrol. Process.,28 (18), 4928-4937, doi:10.1002/hyp.9947.

Alfieri, L., Burek, P., Feyen, L., and Forzieri, G.: Global warming increases the frequency of river floods in Europe, Hydrol. Earth Syst. Sci., 19, 2247–2260, https://doi.org/10.5194/hess-19-2247-2015, 2015.

Arnal, L. et al., 2019. EFAS upgrade for the extended model domain – technical documentation, EUR 29323 EN, Publications Office of the European Union, Luxembourg, 2019, doi: 10.2760/806324, JRC111610

Autorita` di bacino del fiume Po (AdB Po): Progetto di Variante al PAI: mappe della pericolosita` e del rischio di alluvione (in Italian), https://pianoalluvioni.adbpo.it/progetto-esecutivodelleattivita/, accessed on 2020-04-03, 2012.

Dottori, F., Salamon, P., Bianchi, A., Alfieri, L., Hirpa, F.A., Feyen, L., 2016. Development and evaluation of a framework for global flood hazard mapping. Advances in Water Resources 94, 87–102.

Knoben, W. J. M., Freer, J. E., and Woods, R. A.: Technical note: Inherent benchmark or not? Comparing Nash–Sutcliffe and Kling–Gupta efficiency scores, Hydrol. Earth Syst. Sci., 23, 4323–4331, https://doi.org/10.5194/hess-23-4323-2019, 2019.

Krause, P., Boyle, D. P., Bäse, F. Comparison of different efficiency criteria for hydrological model assessment. Advances in Geosciences, European Geosciences Union, 2005, 5, pp.89-97.

Nash, J. E. and Sutcliffe, J. V.: River flow forecasting through conceptual models part I – A discussion of principles, J. Hydrol., 10, 282–290, https://doi.org/10.1016/0022-1694(70)90255-6, 1970.

Trigg, M. et al., 2016. The credibility challenge for global fluvial flood risk analysis. Environ. Res. Lett.11 094014

---

## Author Response (AR3)

**Reply to the Editor**

Dear authors,

many thanks for handing in the revised version of your manuscript on flood hazard maps for Europe. The paper has now been seen by two referees which only raise a few final comments, which should be addressed before the paper can be accepted for publication.

Especially the points of Referee #3, regarding key assumptions of the analysis should be considered since they directly affect the quantitative aspects of the presented data-product.

*We thank the Editor for this additional review effort. Please find below a detailed reply of the points raised by Referee #3. Furthermore we have amended the manuscript as suggested by Referee #4.*

**Reply to Referee #3**

General remarks:

I thank the authors for addressing my comments on the previous version of this manuscript. While I think that most of my comments have been satisfactorily addressed, I would like to insist on two points I find insufficiently addressed despite their importance.

Major points:

1. Distribution choice: Applying a goodness-of-fit test to the distribution used in a frequency analysis is essential. Using a 2-parameter distribution instead of a 3-parameter distribution for parsimony considerations is ok but only if the 2-parameter distribution captures the distribution of the observed floods. I still miss such a goodness-of-fit assessment in the revised version of the manuscript. The single fact that someone else has used a certain distribution for a specific application before does not necessarily make it a good choise for a specific data set.

*We have added a dedicated section in the Appendix (B2) to evaluate and discuss the goodness-of-fit of the Gumbel distribution. We have also included the findings of this additional analysis in the discussion of the validation exercise (lines 425-426, 465-466, 605-606).*

*Appendix B2: "Here we evaluate the performance of the Gumbel distribution in fitting the available reference discharge values (26 annual maxima calculated for all the grid points of the LISFLOOD long-term run). Specifically, we compare the empirical and fitted distributions of streamflow annual maxima using the Cramer-von Mises test (Anderson, 1962), and we calculate the average differences between reference and fitted discharge values. Table B2 summarizes the resulting p-values over the study area. Figure B2 compares empirical and fitted distributions in two locations of the rivers Rhine and Danube. (…) P-values reported in Table B2 suggest a low skill of the fitted Gumbel distributions; however, the resulting*

*uncertainty in the estimates of discharge maxima is generally below 25%, as shown in the examples in Figure B2. This is considered acceptable because the reference discharge maxima are modelled and not observed values. Due to limited sample size, it is not possible to evaluate the extrapolation error for peak flows beyond the available sample; however, previous studies suggested the suitability of the Gumbel distribution. Cunnane (1989) stated that the Gumbel distribution is effective for small sample sizes, whereas the Generalized Extreme Value (GEV) distribution shows a better overall performance if the size is greater than 50. More recently, Papalexiou and Koutsoyiannis (2013) found similar results for extreme precipitation values. In particular, they demonstrated that short record lengths affects the estimation the GEV shape parameter, and thus the choice between a two-parameter (Gumbel) and a three-parameter GEV. Di Baldassarre et al. (2009) observed that the Gumbel distribution might estimate flood extremes with high return periods (e.g. 100-year) with smaller errors than other distributions, if the available sample size is small. Further research could use longer observed streamflow series to compare different extreme value distributions across European regions, similarly to what done by Villarini and Smith (2010) for the eastern United States and Rahman et al. (2013) for Australia."*

2. Design hydrograph construction: The assumption of a triangular hydrograph where the peak occurs in the centre of the hydrograph seems unrealistic as we know that flood hydrographs are asymmetric (more something like 1/3 vs. 2/3 instead of ½ vs. ½). Using a more realistic assumption for the temporal evolution of the event is highly recommended.

*We apologize with the Reviewer the imprecise description provided here. We constructed the design hydrographs following the Chicago Hyetograph methodology, as proposed by Maione et al. (2003)". According to this approach, the hydrograph peak QT is placed in the centre of the hydrograph, while the other values for Qi are sorted alternatively. The resulting hydrograph shape is not symmetric or triangular. Instead, it is fully consistent with all the empirical values of the flow duration curve, taken with a daily step. We revised the text in lines 166-173 accordingly.*

**Additional References**

*Anderson, T. W. (1962). "On the Distribution of the Two-Sample Cramer–von Mises Criterion" (PDF). Annals of Mathematical Statistics. Institute of Mathematical Statistics. 33 (3): 1148–1159. doi:10.1214/aoms/1177704477. ISSN 0003-4851.*

*Cunnane, C. (1989). "Statistical Distributions For Flood Frequency Analysis". Operational Hydrology Report no. 33, World Meteorological Organization*

*Di Baldassarre, G., Laio, F., Montanari, A., 2008. Design flood estimation using model selection criteria. Physics and Chemistry of the Earth 34, 606–611.*

Maione, U., Mignosa, P., & Tomirotti, M. (2003). Regional estimation of synthetic design hydrographs. International Journal of River Basin Management, 1(2), 151-163.

Papalexiou, S. M., and D. Koutsoyiannis (2013), Battle of extreme value distributions: A global survey on extreme daily rainfall, Water Resour. Res.,49, doi:10.1029/2012WR012557

Rahman, A. S., Rahman, A., Zaman, M.A., Haddad, K., Ahsan, A., Imteaz, M., A study on selection of probability distributions for at-site flood frequency analysis in Australia. Nat. Hazards (2013) 69:1803–1813 , doi:10.1007/s11069-013-0775-y

Villarini, G., and J. A. Smith (2010), Flood peak distributions for the eastern United States, Water Resour. Res., 46, W06504, doi:10.1029/2009WR008395.

---

## Author Response (AR4)

**Reply to the Editor**

*We thank the Editor for the positive evaluation of the last revision of our manuscript. Following the requests, we have carefully checked and updated all the links reported in the Data Availability section. Furthermore, we have modified the layout and colours of Figures 4,5,6,C1 to allow a correct interpretation by readers with colour vision deficiencies . Note however that we could not modify the colours of Figure B1 because the figure is taken from a previous work. However, the text describe the relevant elements of this figure.*

*Kind regards*

*Francesco Dottori*